# Fisher meets Feynman: score-based variational inference with a product of experts

**Diana Cai**[1]**, Robert M. Gower**[1]**, David M. Blei**[2]**, Lawrence K. Saul**[1]
[1]Flatiron Institute     [2]Columbia University

## Abstract

We introduce a highly expressive yet distinctly tractable family for black-box variational inference (BBVI). Each member of this family is a weighted product of experts (PoE), and each weighted expert in the product is proportional to a multivariate $t$-distribution. These products of experts can model distributions with skew, heavy tails, and multiple modes, but to use them for BBVI, we must be able to sample from their densities. We show how to do this by reformulating these products of experts as latent variable models with auxiliary Dirichlet random variables. These Dirichlet variables emerge from a Feynman identity, originally developed for loop integrals in quantum field theory, that expresses the product of multiple fractions (or in our case, $t$-distributions) as an integral over the simplex. We leverage this simplicial latent space to draw weighted samples from these products of experts—samples which BBVI then uses to find the PoE that best approximates a target density. Given a collection of experts, we derive an iterative procedure to optimize the exponents that determine their geometric weighting in the PoE. At each iteration, this procedure minimizes a regularized Fisher divergence to match the scores of the variational and target densities at a batch of samples drawn from the current approximation. This minimization reduces to a convex quadratic program, and we prove under general conditions that these updates converge exponentially fast to a near-optimal weighting of experts. We conclude by evaluating this approach on a variety of synthetic and real-world target distributions.

## 1 Introduction

The goal of variational inference (VI) is to approximate an intractable probability density $p$ by the best-matching density $q$ from some simpler parameterized family $\mathcal{Q}$ [4, 23, 53]. VI is typically used when it is difficult to draw samples from $p$, but often there exists a "black-box" way to compute the gradient of $\log p$ (that is, the *score*) at any point in the domain of $\mathbb{R}^D$ [28, 42]. Each such gradient evaluation provides a wealth of information—considerably more than what is provided by a mere sample—and for this reason a growing number of researchers have begun to investigate score-based methods for black-box VI (BBVI) [6, 7, 36, 37]. This direction of research is also motivated by the remarkable successes of score-based methods for generative modeling [18, 19, 47–49].

Despite this allure, score-based BBVI still faces many challenges. There are inherent trade-offs that arise between the expressivity of the variational family $\mathcal{Q}$ and its ease of use. For BBVI it must be tractable to evaluate and draw samples from each $q \in \mathcal{Q}$; it must also be tractable to optimize over $\mathcal{Q}$ and find its best approximation to the target $p$. These trade-offs must be managed by any practitioner of BBVI. At the same time, researchers need more than methods which are merely practical. BBVI replaces an intractable problem in inference by a more tractable one in optimization, but still the challenge remains to prove theoretical guarantees for the optimizations that arise in this framework [3, 11, 24, 57].

39th Conference on Neural Information Processing Systems (NeurIPS 2025).

In this paper we introduce a new family for score-based BBVI that navigates these trade-offs in an appealing fashion. The densities in this family are highly expressive and yet manageably tractable, and we are also able to provide certain theoretical guarantees for the optimizations required for score-based VI. Each density in this family is a *product of experts* (PoE) [16], and each (weighted) expert is proportional to a multivariate $t$-distribution [54] over $\mathbb{R}^D$. In general, it can be challenging to work with products of experts, but for this family we show how to sample from and (in some cases) evaluate their densities—exactly what is needed to use them for VI.

The full potential of this family is unlocked by a Feynman identity [12, 46], originally developed for loop integrals in quantum field theory, that expresses the product of $K$ fractions as an integral over the simplex $\Delta^{K-1}$. In particular, we show that the Feynman identity implies a representation of the product of $t$ densities as a continuous mixture of $t$-distributions. A consequence of this representation is that a PoE in this family is more naturally equipped than a finite mixture model to approximate target densities with a continuum of modes that lie in a convex set. We also leverage this representation to perform two important tasks for BBVI—first, how to draw samples from a PoE in this variational family, and second, how to estimate its normalizing constant. Notably, we transform these problems to samplers and integrals over the simplex $\Delta^{K-1}$ as opposed to all of $\mathbb{R}^D$.

Using these techniques, we then introduce a score-based BBVI algorithm to find the member of this PoE with the minimum Fisher divergence to the target density. To do so, we first generate a large pool of experts (i.e., $t$-distributions with different modes and tails), and then we derive an iterative score-matching procedure to optimize the exponents that determine their geometric weighting in the PoE. In practice this procedure drives many exponents to zero, thus pruning irrelevant experts from the product. More specifically, each iteration updates the expert weights in the PoE by minimizing a regularized Fisher divergence. Score matching is particularly convenient for PoE models because the score is linear in the weights, and the optimization problem reduces to solving a sequence of convex quadratic programs, where each subproblem can be solved efficiently.

In addition, we analyze the convergence of the BBVI algorithm. First, we derive rates of convergence for the expert weights that are iteratively estimated by the algorithm with a finite batch size. We prove that these weights converge exponentially quickly to a neighborhood of an optimal weighting of experts in the PoE, where the size of the neighborhood depends on the amount of misspecification of the variational family. We also demonstrate the benefits of this approach to BBVI empirically on a variety of synthetic and real-world target densities. In particular, we show that with this variational family, we can approximate diverse target densities, including ones that are skewed and heavy-tailed.

## 2 A latent variable model for products of experts

Our goal is to perform score-based BBVI with a particular PoE variational family by minimizing the *Fisher divergence* between a variational density $q(z)$ and a target density $p(z)$ supported on $\mathbb{R}^D$:

$$\mathscr{D}(q; p) = \int \|\nabla \log q(z) - \nabla \log p(z)\|^2 \, q(z) \, dz. \tag{1}$$

Before tackling the broader problem of BBVI with products of experts, we begin by studying these models in their own right, showing how to enable the use of the PoE as a variational family. Consider a (weighted) PoE with the density

$$q(z) = \frac{1}{C_\alpha} \prod_{k=1}^{K} q_k(z)^{\alpha_k}, \tag{2}$$

where the exponents $\{\alpha_k\}_{k=1}^K$ determine the geometric weighting of the nonnegative functions $\{q_k\}_{k=1}^K$ in the product, and $C_\alpha$ is a normalizing constant given by

$$C_\alpha = \int \prod_{k=1}^{K} q_k(z)^{\alpha_k} \, dz. \tag{3}$$

We refer to the function $q_k$ as the $k$th expert in the PoE and to the exponentiated function $q_k^{\alpha_k}$ as the $k$th *weighted* expert. We assume that the exponents (or *weights*) in the PoE are nonnegative (i.e., $\alpha_k \geq 0$), and later we identify further constraints that ensure the integrability of the product in Eq. 3.

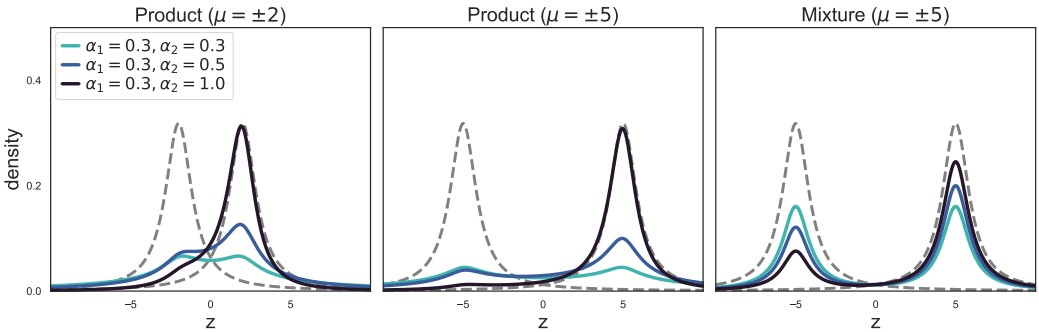

Figure 1: Product vs mixture of two $t$-distributions. The experts (gray dashed curves) all have the same scale of 1. The weights in the mixture (rightmost) are the normalized $\alpha_k$ values.

When the expert functions $q_k$ are Gaussian, the product itself is Gaussian, leading to a closed-form normalizing constant and efficient sampling (see, e.g., Wu and Goodman [56]). However, more generally, it can be difficult to work with products of experts, and in particular to sample from their densities or to estimate their normalizing constants. Because the Fisher divergence in Eq. 1 is generally intractable, we need samples from $q$ to form an empirical estimate of the divergence that can then be minimized. In addition, while the Fisher divergence does not rely on the normalizing constant of $q$, there are many applications that require it; in this work, we use it to compute the Kullback-Leibler (KL) divergence in Section 4.

In this section, we develop a particular family of products of experts, show how to reformulate the models in this family as latent variable models, and then (most critically) leverage the latent space in these models to draw samples from their densities. The ability to sample efficiently from these models will be at the heart of their use for BBVI in Section 3.

## 2.1 Products of multivariate $t$-distributions

We now focus on the parameterized family of products of experts whose weighted experts are proportional to multivariate $t$-distributions over $\mathbb{R}^D$. Specifically, we suppose that

$$q_k(z)^{\alpha_k} = \left[1 + (z-\mu_k)^\top \Lambda_k (z-\mu_k)\right]^{-\alpha_k}, \tag{4}$$

where $\mu_k \in \mathbb{R}^D$ and $\Lambda_k \succeq 0$. Recall that a $t$-distribution is parameterized by its mean $\mu \in \mathbb{R}^D$, inverse scale matrix $\Omega \succ 0$, and degrees of freedom $\nu > 0$, and it has the probability density function

$$\mathcal{T}(z \mid \mu, \Omega, \nu) = \frac{\Gamma(\frac{\nu+D}{2})\,|\Omega|^{1/2}}{\Gamma(\frac{\nu}{2})(\pi\nu)^{D/2}} \left[1 + \tfrac{1}{\nu}(z-\mu)^\top \Omega\,(z-\mu)\right]^{-(\nu+D)/2}, \tag{5}$$

where $\Gamma$ is the gamma function and $|\Omega|$ denotes the determinant of $\Omega$. Thus the $k$th weighted expert in Eq. 4 is proportional to a $t$-distribution with mean $\mu_k$, inverse scale matrix $\Lambda_k$ (when $\Lambda_k \succ 0$), and degrees of freedom $2\alpha_k - D$.

Note that we do *not* require all of the weighted experts in Eq. 4 to define proper densities; instead we allow some of the inverse scale matrices to be *rank-deficient* with $|\Lambda_k| = 0$. Thus these products of experts have more flexibility than finite mixture models whose component densities must be normalizable. The PoE in Eqs. 2 and 4 will be normalizable if $2\sum_k \alpha_k > D$, provided all inverse scale matrices $\Lambda_k$ are full rank. Consequently, we impose this linear sum constraint on the expert weights. If not all inverse scale matrices are positive definite, then integrability must be verified.

We demonstrate the flexibility of this family in Figure 1, where the first two panels show the product of two $t$-distributions that take the form $q(z) \propto \prod_{k=1}^{2} \left[1 + (z-\mu_k)^2\right]^{-\alpha_k}$, where the location parameters are set to either $\mu = \pm 3$ or $\mu = \pm 5$. With only two experts they already parameterize a wide range of behaviors. Finally, we plot the *mixture* of the same two experts as the middle panel, where the mixture weights are formed by normalizing the $\alpha_k$ values. These two panels highlight the difference between products—whose "and" relationship yields larger density values when *all* experts are large—and mixtures—whose "or" relationship yields larger values when *any* one expert is large.

## 2.2 Feynman parameterization for products of experts

We show how to rewrite the PoE in Eq. 2 in a particularly revealing form. To do so, we rely on an identity that expresses a product of positive denominators as an integral over the simplex $\Delta^{K-1} := \{w \in [0, 1]^K : \sum_k w_k = 1\}$. The simplest form of the identity (for two denominators) is straightforward to verify; it states that

$$\frac{1}{A_1 A_2} = \int_0^1 \frac{1}{(A_1 w + A_2(1-w))^2} \, dw \tag{6}$$

for all $A_1, A_2 > 0$. The identity we use generalizes the above to a product of $k$ denominators that are geometrically weighted by exponents $\alpha_k$. In this case, the integral over $[0, 1]$ in Eq. 6 is replaced by an integral over the simplex $\Delta^{K-1}$. In particular, it is also true that

$$\frac{1}{A_1^{\alpha_1} \dots A_K^{\alpha_K}} = \frac{\Gamma(\sum_k \alpha_k)}{\prod_k \Gamma(\alpha_k)} \int_{\Delta^{K-1}} \frac{\prod_k w_k^{\alpha_k - 1}}{(\sum_k w_k A_k)^{\sum_k \alpha_k}} dw, \tag{7}$$

This so-called *Feynman parameterization* is useful to simplify certain loop integrals in quantum field theory [12, 46]. Here we make a telling observation about the terms in Eq. 7 that are independent of the denominators $A_k$; these terms are equal to the probability density function of the Dirichlet distribution. In particular, we can express the Feynman parameterization more compactly as

$$\prod_k A_k^{-\alpha_k} = \mathbb{E}_{w \sim \mathrm{Dir}(\alpha)} \left[ \left( \sum_k w_k A_k \right)^{-\sum_k \alpha_k} \right]. \tag{8}$$

We now use the above identity to reinterpret the PoE introduced in the previous section. First we set $A_k = 1/q_k(z)$ in Eq. 8, where $q_k(z)$ is the expert in Eq. 4. Also, as shorthand, let $\|v\|_\Lambda^2 = v^\top \Lambda v$ denote the quadratic forms that appear in these experts. From these substitutions it follows that

$$\prod_{k=1}^K q_k(z)^{\alpha_k} = \mathbb{E}_{w \sim \mathrm{Dir}(\alpha)} \left[ \left( 1 + \sum_k w_k \|z - \mu_k\|_{\Lambda_k}^2 \right)^{-\sum_k \alpha_k} \right]. \tag{9}$$

We now show that the PoE can be re-expressed as a *continuous* mixture of multivariate $t$-distributions, all of which have the same number of degrees of freedom but differ in other respects; see Appendix C.1 for a detailed derivation. To do so, we define

$$\Lambda(w) = \sum_k w_k \Lambda_k \tag{10}$$

$$\mu(w) = \Lambda(w)^{-1} \sum_k w_k \Lambda_k \mu_k, \tag{11}$$

which can be viewed as location and inverse scale parameters that are continuously indexed by $w \in \Delta^{K-1}$. Now, after expanding the quadratic form in the denominator of Eq. 9 and completing the square, we can use the definitions of $\Lambda(w)$ and $\mu(w)$ to rewrite Eq. 9 as

$$\prod_{k=1}^K q_k(z)^{\alpha_k} = \mathbb{E}_{w \sim \mathrm{Dir}(\alpha)} \left[ \left( 1 + \sum_k w_k \|\mu_k - \mu(w)\|_{\Lambda_k}^2 + \|z - \mu(w)\|_{\Lambda(w)}^2 \right)^{-\sum_k \alpha_k} \right]. \tag{12}$$

Next we supplement Eq. 10 by defining another inverse scale matrix, $\Omega(w)$, that is continuously indexed by $w \in \Delta^{K-1}$ and absorbs the terms in Eq. 12 that are independent of $z$. Let

$$\nu = 2 \sum_k \alpha_k - D, \tag{13}$$

$$\sigma^2(w) = \sum_k w_k \|\mu_k - \mu(w)\|_{\Lambda_k}^2, \tag{14}$$

$$\Omega(w) = \nu \Lambda(w) / (1 + \sigma^2(w)). \tag{15}$$

We can now cast the expectation in Eq. 12 into a more revealing form, arriving at the following representation of the PoE as a continuous mixture of $t$-distributions. This result is obtained by making the substitutions in Eqs. 13 to 15 and appealing to the form of the $t$-distribution in Eq. 5.

---

**Result 2.1** (PoE as continuous mixture of $t$-distributions). Consider the product $\prod_k q_k(z)^{\alpha_k}$ with $t$-distributed experts (Eq. 4), and let $\nu = 2 \sum_k \alpha_k - D$. The product can be computed as

$$\prod_{k=1}^K q_k(z)^{\alpha_k} = \mathbb{E}_{w \sim \mathrm{Dir}(\alpha)} \left[ (1 + \sigma^2(w))^{-\frac{\nu+D}{2}} \left( 1 + \frac{1}{\nu} \|z - \mu(w)\|_{\Omega(w)}^2 \right)^{-\frac{\nu+D}{2}} \right]. \tag{16}$$

Thus the PoE can be viewed as a *continuous mixture of $t$-distributions* in Eq. 5 with $\nu$ degrees of freedom, location parameters $\mu(w)$, and inverse scale matrices $\Omega(w)$, where $w \in \Delta^{K-1}$.

---

This result suggests that this type of PoE may be well-suited for distributions with a continuum of modes that lie in a convex set. Any such regions of high probability will be more naturally modeled by a continuous mixture than a finite one. As an aside, we note that the $t$-distribution can itself be written as a (continuous) scale mixture of Gaussians [1]. By extension, the above result shows that a product of $t$-distributed experts can be written even more generally as a continuous *location-and-scale* mixture of Gaussians. This correspondence provides further motivation for their use in VI.

### 2.3 Joint density and latent variable model

There are many useful results that follow from the Feynman parameterization. One such result arises when revisiting the computation of the PoE's normalizing constant $C_\alpha$ in Eq. 3. While $C_\alpha$ is initially expressed as an integral over $\mathbb{R}^D$, our goal is to re-express it as a (potentially simpler) integral over the simplex $\Delta^{K-1}$. Substituting Eq. 16 into Eq. 3, we find that

$$C_\alpha = \mathbb{E}_{w \sim \text{Dir}(\alpha)} \left[ \left(1+\sigma^2(w)\right)^{-\frac{\nu+D}{2}} \int dz \left(1 + \tfrac{1}{\nu}\|z-\mu(w)\|^2_{\Omega(w)}\right)^{-\frac{\nu+D}{2}} \right], \tag{17}$$

where we have applied Fubini's theorem to move the integral over $z$ inside the expectation over $w$. We can now perform the integral over $z$, as it is given exactly by the normalizing constant in Eq. 5 for a $t$-distribution with inverse scale matrix $\Omega(w)$ and $\nu$ degrees of freedom. In this way we obtain the following result.

---

**Result 2.2** (Normalizing constant of PoE). Consider the PoE with the $t$-distributed experts in Eq. 4, and let $\nu = 2\sum_k \alpha_k - D$. The normalizing constant of this PoE can alternately be computed as

$$C_\alpha = \int \prod_{k=1}^K q_k(z)^{\alpha_k} \, dz = \mathbb{E}_{w \sim \text{Dir}(\alpha)} \left[ |\Omega(w)|^{-\frac{1}{2}} \left(1+\sigma^2(w)\right)^{-\frac{\nu+D}{2}} \right] \cdot \frac{\Gamma(\frac{\nu}{2})(\pi\nu)^{\frac{D}{2}}}{\Gamma(\frac{\nu+D}{2})} \tag{18}$$

---

Next we use the Feynman parameterization to show how to sample from $q$. In particular, we construct a joint density $q(w, z)$ that yields the desired marginal $q(z)$ in Eq. 2, and then we generate samples from the joint $q(w, z)$. Combining Result 2.1 and Result 2.2, we find

$$q(w, z) = \frac{C_\alpha(w)}{C_\alpha} \, \text{Dir}(w \mid \alpha) \, \mathcal{T}(z \mid \mu(w), \Omega(w), \nu), \tag{19}$$

where the new leading factor $C_\alpha(w)$ in the numerator is used to account for all the terms in Eq. 17 that are not absorbed by the (normalized) Dirichlet and $t$-distributions—namely,

$$C_\alpha(w) := \left[ |\Omega(w)| \left(1+\sigma^2(w)\right)^{\nu+D} \right]^{-\frac{1}{2}} \frac{\Gamma(\frac{\nu}{2})(\pi\nu)^{\frac{D}{2}}}{\Gamma(\frac{\nu+D}{2})}. \tag{20}$$

It is then straightforward to verify that $\int q(w, z)\,dw = q(z)$, leading to a natural auxiliary-variable sampling procedure. Indeed, we can interpret this joint density as a latent variable model and use its marginal and conditional densities to draw samples from the PoE. Marginalizing over $z$ in Eq. 19, we find that $q(w) = \int q(w, z)\,dz = \frac{C_\alpha(w)}{C_\alpha} \text{Dir}(w \mid \alpha)$, where $C_\alpha(w)$ is given by Eq. 20. Likewise, we recover the $t$-distribution for the conditional density $q(z|w)$ upon dividing the joint in Eq. 19 by this result. In sum we have shown the following.

---

**Result 2.3** (Latent variable model for PoE). Consider the PoE with the $t$-distributed experts in Eq. 4. We can draw samples from the PoE by sampling from the generative model

$$w_b \sim \frac{C_\alpha(w)}{C_\alpha} \, \text{Dir}(w \mid \alpha), \tag{21}$$

$$z_b \mid w_b \sim \mathcal{T}(z \mid \mu(w_b), \Omega(w_b), \nu), \tag{22}$$

where $w_b$ lies in the simplex, the $w_b$-dependent terms on the right are given by Eqs. 10, 11, 13 to 15 and 20, and $z_b$ is conditionally $t$-distributed given $w_b$.

---

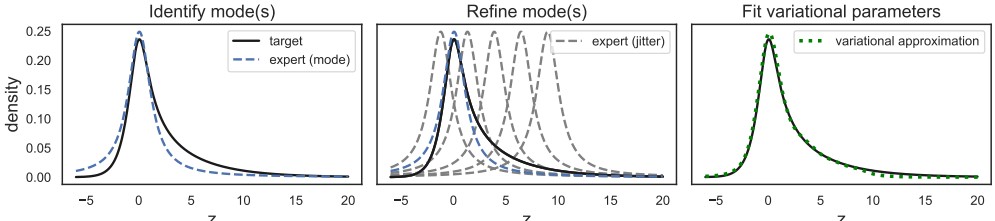

Figure 2: To select experts, we identify each mode and then add more experts to refine the fit.

The above result is not yet a practical recipe for sampling from a PoE. The difficulty lies in the first step: it is not straightforward to sample from $q(w)$ in Eq. 21 due to its leading dependence on $C_\alpha(w)$. But we can circumvent this difficulty by using the Dirichlet distribution that appears in $q(w)$ as a proposal distribution for importance sampling.

Suppose we wish to estimate an expected value $\mathbb{E}_{q(z)}[h(z)]$. We can draw a batch of samples

$$w_b \sim \text{Dir}(w \,|\, \alpha), \tag{23}$$
$$z_b \,|\, w_b \sim \mathcal{T}(z \,|\, \mu(w_b), \Omega(w_b), \nu). \tag{24}$$

from the Dirichlet and $t$-distributions in Eqs. 21 and 22 and weight these samples by $C_\alpha(w_b)$ in the calculation of the expected value. In this way we obtain the estimate

$$\mathbb{E}_{q(z)}[h(z)] \approx \frac{\sum_b C_\alpha(w_b)\, h(z_b)}{\sum_b C_\alpha(w_b)}. \tag{25}$$

## 3 Score-based VI with products of experts

In this section we show how to approximate a target density $p$ by a PoE with $t$-distributed experts. To do so, we must specify the number of experts, the parameters of their $t$-distributions, and the exponents that determine their geometric weighting in the PoE. We discuss these problems in turn.

### 3.1 Selecting the experts

Of the many ways to select experts, we seek a simple heuristic that works in practice and avoids a complicated, coupled optimization over the expert parameters $(\mu_k, \Lambda_k)$ and weights $\alpha_k$ in Eq. 4. Our basic strategy is to generate a large, oversaturated pool of experts whose means $\mu_k$ are concentrated near the modes of the target density. We shall see in later sections that poorly situated (and hence irrelevant) experts are efficiently pruned by the procedure for learning the weights $\alpha_k$.

Our strategy for selecting experts has three steps. The first is to locate the modes of the target density by hill-climbing in $\log p(z)$ from randomly chosen starts; if the target density is a Bayesian posterior, then we can choose these starts by sampling from the corresponding prior. The second step is to place an expert at each mode: the expert's mean $\mu_k$ and inverse scale parameter $\Lambda_k$ to match the mode's location and curvature. The third step is to place additional experts nearby so that the weighted PoE can better model the shape (e.g., skew, kurtosis) of each mode: once a new location $\mu_k$ is chosen, the corresponding inverse scale $\Lambda_k$ is set to the (negative) Hessian at $\mu_k$ projected to the cone of positive semidefinite matrices. For further details of this step and a discussion of expert placing cost, see Appendix D.1. Figure 2 illustrates the intuition behind this strategy for a target density with one mode. Overall we found this strategy to be quite effective in conjunction with a score-based procedure for optimizing the expert weights in the PoE. We describe this score-based procedure next.

### 3.2 Weighting the experts

Given experts $q_k$ in the form of Eq. 4, we now consider how to form the weighted PoE in Eq. 2 that best approximates the target density $p$. Specifically, for $q(z) \propto \prod_k q_k(z)^{\alpha_k}$, we seek the weights $\{\alpha_k\}_{k=1}^K$ that minimize the *Fisher divergence* between $q$ and $p$, as defined in Eq. 1. Since the PoE in Eq. 2 has support on $\mathbb{R}^D$, this divergence vanishes only when $q = p$.

Though we cannot minimize Eq. 1 directly, in the spirit of BBVI, we can instead attempt to minimize an empirical estimate of the Fisher divergence, one that is based on drawing samples from $q$. A further simplification is achieved by decoupling the procedures for sampling from $q$ and optimizing $q$. To do so, we iteratively minimize a closely related objective. In particular, let $\alpha^{(t)} \in \mathbb{R}^K$ be the expert weights at the $t^{\text{th}}$ iteration, and let $q(z|\alpha)$ denote the density of the PoE with expert weights $\alpha$. Rather than minimizing Eq. 1 directly, at the $t^{\text{th}}$ iteration we instead solve the simpler problem

$$\alpha^{(t+1)} = \underset{\alpha \in \mathscr{C}}{\operatorname{argmin}} \left\{ \int \left\| \nabla \log q(z|\alpha) - \nabla \log p(z) \right\|^2 q(z|\alpha^{(t)}) \, dz \;+\; \tfrac{1}{\eta_t} \left\| \alpha - \alpha^{(t)} \right\|^2 \right\}, \quad (26)$$

where $\eta_t > 0$ is a learning rate and the domain $\mathscr{C}$ of the optimization constrains the expert weights to define a normalizable PoE. Note that this update minimizes a *biased* estimate of the Fisher divergence; the estimate is biased because in the first term of the objective the expectation is performed with respect to $q(z|\alpha^{(t)})$ instead of $q(z|\alpha)$. But at the same time, the update attempts to compensate for this bias by penalizing solutions that move too far from one iteration to the next; this penalty is enforced by the regularizer in the second term of the objective. This is the same intuition that is behind a recently proposed "batch-and-match" algorithm for Gaussian BBVI [7].

The rest of this section fleshes out this iterative procedure and highlights its three main advantages. First, when $q$ is a weighted PoE with $t$-distributed experts, we can use samples to compute an empirical estimate of the objective in Eq. 26. Second, at each iteration, we can minimize this empirical estimate by solving a strongly convex quadratic program. Third, this iterative procedure provably converges under fairly general conditions to a neighborhood of an optimally weighted PoE.

We now construct an empirical estimate $\widehat{\mathcal{E}}_t(\alpha)$ for the objective in Eq. 26 at the $t^{\text{th}}$ iteration. Using the latent variable model in Eqs. 21 and 22, we generate a batch of $B$ weighted samples $\{(w_b, z_b)\}_{b=1}^B$ from the PoE with expert weights $\alpha^{(t)}$. From these samples, we construct the empirical estimate

$$\widehat{\mathcal{E}}_t(\alpha) = \tfrac{1}{B} \sum_{b=1}^B \pi_b \left\| \nabla \log q(z_b|\alpha) - \nabla \log p(z_b) \right\|^2 + \tfrac{1}{\eta_t} \left\| \alpha - \alpha^{(t)} \right\|^2, \quad (27)$$

where $\pi_b \propto C_{\alpha^{(t)}}(w_b)$ is the importance weight of the $b^{\text{th}}$ sample from Eq. 25. Note how by sampling the PoE with weights $\alpha^{(t)}$, we have decoupled these samples from the optimization over $\alpha$ in Eq. 26.

Next we show that the empirical estimate $\widehat{\mathcal{E}}_t(\alpha)$ is minimized by solving a convex quadratic program in the expert weights $\alpha$. First, we observe that for any PoE, as defined by $q(z|\alpha) \propto \prod_k q_k(z)^{\alpha_k}$ in Eq. 2, *the score is linear in the weights of its experts*: namely, $\nabla \log q(z|\alpha) = \sum_k \alpha_k \nabla \log q_k(z)$. We make this explicit for the $t$-distributed experts in Eq. 4 by writing

$$\nabla \log q(z_b|\alpha) = Q_b \alpha, \quad (28)$$

where $Q_b$ is the $D \times K$ matrix whose $k^{\text{th}}$ column is given by $\nabla \log q_k(z_b) = -2q_k(z_b)\Lambda_k(z_b - \mu_k)$. From the linearity of the scores, it follows at once that $\widehat{\mathcal{E}}_t(\alpha)$ in Eq. 27 is quadratic in the expert weights $\alpha$. In particular, we can rewrite Eq. 27 as

$$\widehat{\mathcal{E}}_t(\alpha) = \alpha^\top \left[ \tfrac{1}{B} \textstyle\sum_b \pi_b Q_b^\top Q_b + \tfrac{1}{\eta_t} I \right] \alpha - 2 \left[ \tfrac{1}{B} \textstyle\sum_b \pi_b Q_b^\top \nabla \log p(z_b) + \tfrac{1}{\eta_t} \right]^\top \alpha \;+\; \widehat{\mathcal{E}}_t(0), \quad (29)$$

and from the above, we also see that $\widehat{\mathcal{E}}_t(\alpha)$ is strongly convex in $\alpha$ for all $\eta_t \in (0, \infty)$. The expert weights are updated by minimizing this objective subject to a constraint that the newly weighted PoE is normalizable. To satisfy the parameter constraints of the PoE, we define the constraint set, such that $\alpha \in \mathscr{C}$, as

$$\mathscr{C} = \left\{ \alpha \in \mathbb{R}^K : \alpha_1 \geq 0, \, \alpha_2 \geq 0, \, \ldots, \alpha_k \geq 0, \, \textstyle\sum_k \alpha_k \geq \tfrac{D}{2} + \varepsilon \right\}, \quad (30)$$

where $\varepsilon > 0$ is a slack variable added to ensure that the product of experts $\prod_k q_k(z)^{\alpha_k}$ is integrable. Eq. 30 defines a convex set, and hence the overall optimization is convex; in particular, it is a nonnegative least squares (NNLS) problem with linear constraints, for which there exist many efficient solvers [29, Ch. 23]. For our purposes, it is also interesting that problems in NNLS often yield *sparse* solutions where multiple constraints are active. In our setting, these are solutions in which irrelevant experts are assigned zero weights and do not contribute to the density of the PoE.

**Algorithm 1** Score-based VI for PoE: learning the weights $\alpha$

---

1: **Input**: initial weights $\alpha^{(0)} \in \mathscr{C}$, target score function $\nabla \log p$, experts $q_k(z)$, learning rates $\eta_t > 0$.
2: **for** $t = 1, \ldots, T$: **do**
3:     **for** $b = 1, \ldots, B$: **do**
4:         Sample $w_b \sim \text{Dir}(w \mid \alpha^{(t)})$ and $z_b \sim \mathcal{T}(z \mid \mu(w_b), \Omega(w_b), \nu^{(t)})$.
5:         Compute importance weight $\pi_b \propto C_{\alpha^{(t)}}(w_b)$ and target score $g_b = \nabla \log p(z_b)$.
6:         Compute matrix $Q_b \in \mathbb{R}^{D \times K}$ whose $k$th column is equal to $\nabla \log q_k(z_b)$.
7:     **end for**
8:     Compute $G_t = \frac{1}{B} \sum_{b=1}^B \pi_b Q_b^\top Q_b + \frac{1}{\eta_t} I$ and $h_t = \frac{1}{B} \sum_{b=1}^B \pi_b Q_b^\top g_b + \frac{1}{\eta_t} \alpha^{(t)}$.
9:     Update expert weights: $\alpha^{(t+1)} = \text{argmin}_{\alpha \in \mathscr{C}} \left[ \frac{1}{2} \alpha^\top G_t \alpha - h_t^\top \alpha \right]$.
10: **end for**
11: **Output**: PoE weights $\alpha^{(T)} \in \mathscr{C}$ (Eq. 30)

---

### 3.3 Convergence theorem

We summarize the overall iterative procedure for learning expert weights in Algorithm 1. To prove convergence we make some basic assumptions about the Fisher divergence in Eq. 1 and its empirical estimate at each iteration of Algorithm 1. To state these assumptions, let

$$\widehat{\mathscr{D}}_t(\alpha) = \tfrac{1}{B} \textstyle\sum_b \pi_b \big\| \nabla \log q(z_b|\alpha) - \nabla \log p(z_b) \big\|^2 \tag{31}$$

denote the first term in Eq. 27, and let $\nabla \widehat{\mathscr{D}}_t$ and $\mathcal{H}[\widehat{\mathscr{D}}_t]$ denote the gradient and Hessian of this term with respect to the expert weights $\alpha$. With these definitions, we can state the following theorem.

---

**Theorem 3.1.** Suppose that $\mathscr{D}(q; p)$ in Eq. 1 is minimized by a unique $\alpha^* \in \mathcal{C}$, and also that for all $t \geq 0$ there exists some $\delta \geq 0$ such that $\mathbb{E} \big\| \frac{1}{2} \nabla \widehat{\mathscr{D}}_t(\alpha^*) \big\| \leq \delta$ and some $\lambda > 0$ such that $\mathcal{H}[\widehat{\mathscr{D}}_t] \succeq \lambda I$ almost surely. Then for constant learning rates $\eta_t \equiv \eta$, the expected error of the iterates satisfies

$$\mathbb{E} \big\| \alpha^{(t)} - \alpha^* \big\| \leq \left( \tfrac{1}{1+\eta\lambda} \right)^t \big\| \alpha^{(0)} - \alpha^* \big\| + \tfrac{\delta}{\lambda}. \tag{32}$$

---

This result ensures that the iterates $\alpha^{(t)}$ of Algorithm 1 converge, in expectation, to a neighborhood around the optimal expert weights $\alpha^*$, with the error decaying at a linear (geometric) rate. The proof relies on a few key ideas, most notably that (i) the constrained least-squares problem in Eq. C.6 can be solved by projecting its unconstrained solution onto $\mathcal{C}$, and (ii) this projection is with respect to an induced Mahalanobis norm, and it is nonexpansive [2, Proposition 4.8]. See Appendix E for a complete proof.

The bound in Eq. 32 separates two effects—the transient error, which shrinks exponentially fast with $t$ and depends on the *strong convexity parameter* $\lambda$, and the asymptotic error floor $\frac{\delta}{\lambda}$, which depends also on the *misspecification* parameter $\delta$. We briefly sketch the intuition behind these terms.

First, the error floor is proportional to the misspecification parameter $\delta$. Let $q^*$ denote the optimally weighted PoE. On one hand, if $p \in \mathcal{Q}$, then $q^* = p$ and $\widehat{\mathscr{D}}_t(\alpha^*) = \|\nabla \widehat{\mathscr{D}}_t(\alpha^*)\| = 0$ for all $t$; we see in this case that $\delta = 0$. On the other hand, if $p \notin \mathcal{Q}$, then $q^* \neq p$. In this case we expect that the stochastic gradients will have small norms at $\alpha^*$ (and thus $\delta$ will be small) whenever $\mathscr{D}(q^*; p)$ itself is small. In the right panel of Figure D.2 we highlight on one experiment with an sinh-arcsinh target, that for constant or decreasing step size schedules, the error floor does not go below $10^{-1}$. However, on the left panel of Figure D.2, we do show that this error floor improves as we increase the batch size. See Appendix D.2 and Appendix D.3 for details.

Second, the transient error depends on the assumption that $\widehat{\mathscr{D}}_t(\alpha)$ is $\lambda$-strongly convex for all $t$. Observe from Eq. C.6 that $\mathcal{H}[\widehat{\mathscr{D}}_t] = \frac{1}{B} \sum_{b=1}^B \pi_b Q_b^\top Q_b$; i.e., the Hessian is a sum of $B$ positive semidefinite matrices. Thus for larger batch sizes, the assumption of strong convexity is increasingly likely to be satisfied. We confirm this intuition in Figure E.1, where we show that as we increase the batch size, the eigenvalues of the Hessian also increase and are bounded away from zero. We only encountered one indefinite Hessian in our experiments; this occurred when the number of experts ($K = 100$) was much larger than the batch size ($B = 10$).

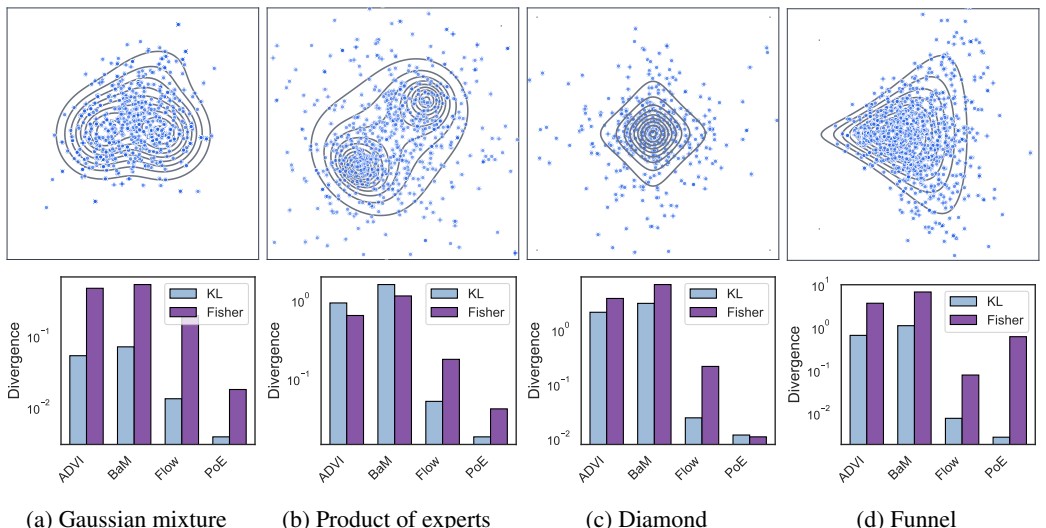

Figure 3: Synthetic 2D targets. **Top:** Gray contours represent the target, and blue points represent samples from the fitted PoE. **Bottom:** The KL and Fisher divergences of each method.

## 4 Experiments

In this section, we compare VI with this PoE family to Gaussian BBVI (with ADVI [28] and BaM [7]) and normalizing flow-based VI [43]. In Appendix F, we provide more details on these experiments and also results on several additional examples. We also present additional results for estimating PoE normalizing constants and evaluating sampling (Appendix F.1) and expert placement (Appendix D.2).

### 4.1 Synthetic 2D targets

We first consider several synthetic 2D target distributions: 1) Mixture of Gaussians, 2) Product of $t$-distributions, 3) Diamond, 4) Funnel. We report Monte Carlo estimates of two training-objective-agnostic metrics, $\mathrm{KL}(p; q)$ and $\mathbb{E}_p[\|\nabla \log q - \nabla \log p\|^2]$, using 1000 samples from $p$. (The VI methods minimize the expectations with respect to $q$.) We visualize the PoE samples by first drawing a weighted sample and then resampling the generated samples according to their normalized weights. For full details on each target, see Appendix F.

Figure 3 shows each target distribution's contours (gray curves) overlaid with samples from the fitted PoE (blue points). In these examples, the PoE achieves substantially lower values for both divergence metrics than the Gaussian approximation, due to its ability to better capture the tails of the target distributions. The normalizing flow also provides a substantial improvement over the Gaussian fit. Notably, for heavy-tailed (product of experts and diamond) targets, the PoE outperforms the flow-based family as well, yielding lower divergence in these cases.

### 4.2 Sinh-arcsinh target

We now evaluate a higher-dimensional synthetic target distribution with skew and heavy tails. The sinh-arcsinh distribution [21, 22] generalizes the Gaussian distribution with additional parameters $\varepsilon \in \mathbb{R}^D$ and $\tau \in \mathbb{R}^D_{++}$ that control the skewness and the tail-weight. It transforms a Gaussian draw $\tilde{z} \sim N(0, \Sigma)$ coordinate-wise via $z_d = \sinh((\sinh^{-1}(\tilde{z}_d) + \varepsilon_d)/\tau_d)$. We construct the target density in $D = 50$ dimensions so that it is positively skewed ($\varepsilon = 0.3$) and heavy-tailed ($\tau_d = 0.7$).

In Figure 4, we show the KL and Fisher divergences between all approaches, and we found that the PoE had the lowest divergence for both metrics (left). We also plotted the divergences against the number of gradient evaluations for the iterative parameter fitting of the PoE and the normalizing flow. Note that this plot does not show the initial startup costs of each method (selecting the experts for the PoE and selecting a learning rate for the flow), which are not the dominating cost. In addition, this plot only shows the first $15 \times 10^4$ gradient evaluations. For both divergences, the values decrease slowly for the flow model; the final reported value in the bar graph is after $10^7$ gradient evaluations.

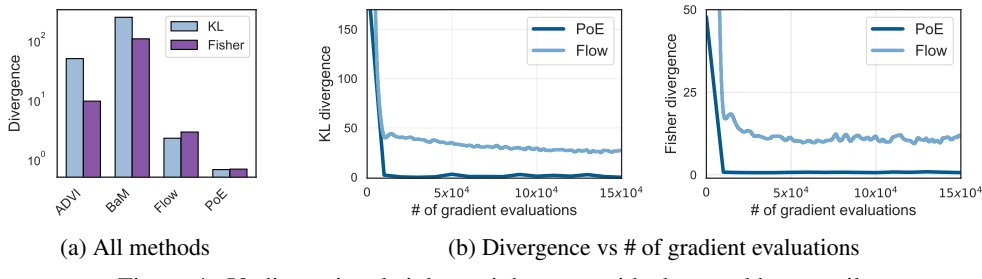

(a) All methods          (b) Divergence vs # of gradient evaluations

Figure 4: 50-dimensional sinh-arcsinh target with skew and heavy tails.

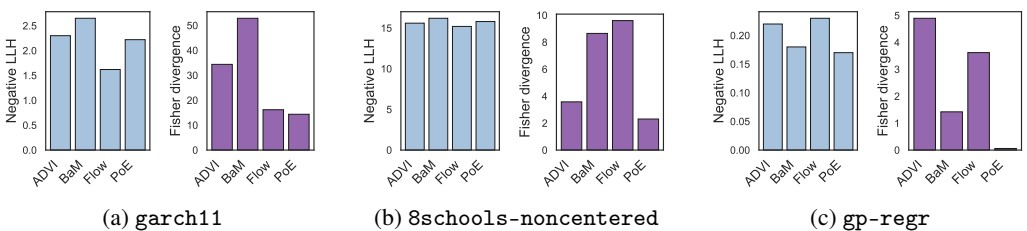

(a) `garch11`       (b) `8schools-noncentered`       (c) `gp-regr`

Figure 5: `posteriordb` targets that highlight a range of non-Gaussian posterior properties.

### 4.3 Posterior inference problems

Next we study several posterior inference test problems from `posteriordb` [32, 33]. For each target, we use reference samples computed using Stan (drawn via Hamiltonian Monte Carlo) [9]. Because we do not have access to the normalized target density $p(z)$, instead of reporting $\text{KL}(p; q)$, we report the negative expected log likelihood $\text{Neg-LLH}(p; q) = -\mathbb{E}_p[\log(q)]$. We again also consider the Fisher divergence $F(p; q)$, which does not require normalizing constants. All expectations are estimated using the reference samples from $p$.

In Figure 5, we highlight three particular examples from `posteriordb`. First, we consider `garch11`: this model has (half) uniform priors that skew the posterior when the support is transformed to $\mathbb{R}^D$. The skew is not modeled by Gaussian VI, but it is present in the PoE and flow-based approximations. Next, we consider `8schools-noncentered`, which exhibits skew and a heavy-tailed component introduced by a half-Cauchy$(0, 5)$ prior. These properties are best modeled by the variational PoE. Finally, we study a light-tailed example in `gp-regr`, where Gaussian BBVI performs well due the near symmetry of the posterior. Nevertheless, despite its heavy tails, the product of $t$-distributions still outperforms both Gaussian BBVI and the normalizing flow (with a Gaussian base distribution) on this example. This example illustrates how the expert weights $\alpha$ can be optimized to match different tails via the value of $\nu = 2 \sum_k \alpha_k - D$ in the mixture representation of Result 2.1.

## 5 Discussion of contributions, limitations, and future work

We have shown that VI with products of $t$-experts can model a variety of target densities. A key technical insight, via a Feynman identity, was to represent each PoE as a continuous mixture indexed by a Dirichlet variable. We then used this representation to sample from the PoE, a core requirement for BBVI. To optimize the expert weights, we developed a score-based VI algorithm that solves a sequence of convex quadratic programs, and we proved that its iterates converge exponentially to a neighborhood that depends on the degree of misspecification of the variational family.

While our algorithm learns the expert weights, it is limited by relying on a fixed collection of experts selected upfront. In practice, the quality of the approximation depends heavily on how these experts are chosen. While our current heuristic overspecifies the number of experts and prunes away the ineffectual ones, further gains could be achieved with a more refined approach—for instance, a boosting-style approach [15, 35] that sequentially adds experts based on (say) score mismatch.

Several promising directions remain. First, the Feynman identity may have broader implications: for example, by providing a semi-analytic procedure to estimate the normalizing constant, it may open a new door to generative models with products of $t$-experts [26, 54]. Second, the PoE construction may be useful in certain sampling methods as an initialization or proposal scheme. Finally, as mentioned above, by choosing experts more carefully we can hope to accelerate every aspect of our approach.

## Acknowledgements

D.B. was supported by NSF IIS-2127869, NSF DMS-2311108, ONR N000142412243, and the Simons Foundation.

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

# Appendix

## A  Overview

In this supplementary material we provide additional details on several aspects of the paper. We start by providing a discussion of the broader VI literature in Appendix B, focusing on VI methods that employ expressive variational families. Then, in Appendix C we provide additional derivations and details about the Feynman parameterization. In particular, we provide a full derivation of the continuous mixture representation of the PoE presented in Result 2.1, along with an expanded discussion of the normalizing constant. Next, in Appendix D we present one heuristic used for selecting the experts, along with an empirical study of the heuristic. In Appendix E, we provide the proof for the convergence theorem, which is supplemented with an empirical investigation. Finally, in Appendix F, we present additional empirical results. In particular, we first study the properties of the PoE family, including the estimation of the normalizing constant and the sampling procedure. We then provide details about the setup of the experiments in Section 4, including more details about the tuning of the baselines and the implementation of the PoE score matching algorithm. Next, we provide additional plots and analysis for the experiments in Section 4.

## B  Related work

**Mixture modeling.**  There have been many efforts to extend VI beyond families of factorized or multivariate Gaussian distributions. One natural extension is to choose the approximating distribution from a parameterized family of mixture models (i.e., a mixture of experts), and in this line of work, it has been most common to study Gaussian mixture models. Gershman et al. [14] introduced an infinite Gaussian mixture model that draws inspiration from kernel density estimation. In a different approach, Morningstar et al. [38] integrated mixture families into ADVI using stratified sampling and optimizing a tighter evidence lower bound via gradient descent. More recently, Xu et al. [58] combined the variational families of mixture models and normalizing flows, establishing MCMC-like convergence guarantees.

**Boosting VI approaches.**  Mixture models have also been used in boosting strategies for VI. In this approach, the variational approximation is iteratively improved by adding new components to the mixture. Guo et al. [15] and Miller et al. [35] proposed greedy algorithms that incrementally add Gaussian components to reduce the KL divergence. Locatello et al. [31] later reinterpreted this boosting procedure as a Frank–Wolfe algorithm, deriving explicit convergence guarantees. Campbell and Li [8] replaced the KL divergence with the Hellinger distance; with this objective, the optimization of mixture weights reduces to a problem in nonnegative least squares that is similar in flavor to the optimization of PoE weights in our work.

**Basis-set variational families.**  A related line of work has explored variational approximations that are parameterized by linear combinations of basis functions. For example, in EigenVI [6] the variational approximation is parameterized as the square of a linear combination of orthogonal functions, and the best approximation is found by minimizing a Fisher divergence—an optimization that reduces in turn to a minimum-eigenvalue problem. However, EigenVI scales exponentially with the latent dimension, limiting its applicability. Shao et al. [45] used spline-based approximations within an ADVI framework, increasing the flexibility of the variational approximation.

**Normalizing flow-based families.**  Another line of work has explored the use of normalizing flows [27, 43], where the variational approximation is parameterized by an invertible neural network and found by optimizing the ELBO using the reparameterization trick. Jaini et al. [20] and Liang et al. [30] introduced tail-adaptive components into flow-based models in order to better approximate non-Gaussian posteriors with heavy tails. Normalizing flows with these architectures have enjoyed empirical success, but they can be difficult to optimize and generally lack convergence guarantees when they are used for VI.

**Implicit variational inference.**  Implicit VI [51, 52, 59] defines variational families using a continuous mixture of a kernel. The product of experts we consider can be viewed as fitting in the framework of implicit VI with a particular kernel ($t$-distribution) and mixing measure (Dirichlet) due to Result 2.1; the PoE here is a particular function of the auxiliary variables. On the other hand, the typical implicit VI setting specifies a neural network with a mixing distribution over the parameters

of that neural network. In this special case of continuously parameterized models, we are able to identify tools that lead to both an expressive family of densities but also a tractable normalizing constant via the Feyman representation.

**Products of experts.** There has been relatively less work on products of experts for VI. In part they have been underexplored because it is generally intractable to compute their normalizing constants, and in one way or another these normalizing constants enter into the optimizations for ELBO-based BBVI. Products of experts (including $t$-distributed experts [54]) have been used for generative modeling [16, 17], but in these studies the learning was finessed by computing gradients of the contrastive divergence instead of the KL divergence. Products of experts can also be viewed as a special case of energy-based models. The latter have been used for VI [60], but in this context they are typically paired with Markov chain Monte Carlo (MCMC) methods.

## C  Latent variable model for products of experts: additional details

### C.1  Feynman parameterization of product of $t$-experts (Result 2.1)

We now provide a full derivation of the Feynman parameterization of the PoE with $t$-experts. In this case, the denominators have the form

$$A_k := 1 + (z - \mu_k)^\top \Lambda_k (z - \mu_k). \tag{C.1}$$

Now we substitute the denominators $A_k$ into the Feynman parameterization in Eq. 7:

$$\prod_{k=1}^{K} q_k(z)^{\alpha_k} = \prod_{k=1}^{K} \frac{1}{\big[\underbrace{1 + (z - \mu_k)^\top \Lambda_k (z - \mu_k)}_{=A_k}\big]^{\alpha_k}} \tag{C.2}$$

$$= \frac{\Gamma(\sum_k \alpha_k)}{\prod_k \Gamma(\alpha_k)} \int_{\Delta^{K-1}} \frac{\prod_k w_k^{\alpha_k - 1}}{(\sum_k w_k[1 + (z - \mu_k)^\top \Lambda_k (z - \mu_k)])^{\sum_k \alpha_k}} dw. \tag{C.3}$$

Now we will rewrite the denominator in Eq. C.3 in a convenient form by expanding the terms and completing the square. In particular, the sum in the denominator can be written as

$$\sum_k w_k \left[ 1 + (z - \mu_k)^\top \Lambda_k (z - \mu_k) \right] \tag{C.4}$$

$$= \sum_k w_k + \sum_k w_k (z - \mu_k)^\top \Lambda_k (z - \mu_k) \tag{C.5}$$

$$= 1 + z^\top \left( \sum_k w_k \Lambda_k \right) z - 2 z^\top \left( \sum_k w_k \Lambda_k \mu_k \right) + \sum_k w_k \mu_k^\top \Lambda_k \mu_k, \tag{C.6}$$

where in the last line we have isolated the terms that are quadratic and linear in $z$. Our next step is to complete the square, noting that

$$z^\top A z - 2 z^\top b = \left\| z - A^{-1} b \right\|_A^2 - b^\top A^{-1} b \tag{C.7}$$

for any invertible, symmetric matrix $A$ and vector $b$. Next we identify $A = \sum_k w_k \Lambda_k$ and $b = \sum_k w_k \Lambda_k \mu_k$ from the quadratic and linear terms in Eq. C.6. Completing the square in this way, we can write

$$\sum_k w_k \left[ 1 + (z - \mu_k)^\top \Lambda_k (z - \mu_k) \right] = 1 + \left\| z - \mu(w) \right\|_{\Lambda(w)}^2 + \sigma^2(w), \tag{C.8}$$

where we have defined

$$\Lambda(w) = \sum_k w_k \Lambda_k \tag{C.9}$$

$$\mu(w) = \Lambda^{-1}(w) \sum_k w_k \Lambda_k \mu_k \tag{C.10}$$

$$\sigma^2(w) = -\mu(w)^\top \Lambda(w) \mu(w) + \sum_k w_k \mu_k^\top \Lambda_k \mu_k. \tag{C.11}$$

Next we observe that the expression for $\sigma^2(w)$ in Eq. C.11 can be written more simply as

$$\sigma^2(w) = \sum_k w_k \|\mu_k - \mu(w)\|^2_{\Lambda_k},$$
(C.12)

so that Eq. C.8 reproduces the earlier result for this denominator given by Eq. 12.

Thus, after completing the square and rearranging terms, the product of experts can be written as:

$$\prod_{k=1}^{K} q_k(z)^{\alpha_k} = \mathbb{E}_{w \sim \text{Dir}(\alpha)} \left[ \frac{1}{(\sum_k w_k[1 + (z - \mu_k)^\top \Lambda_k(z - \mu_k)])^{\sum_k \alpha_k}} \right]$$
(C.13)

$$= \mathbb{E}_{w \sim \text{Dir}(\alpha)} \left[ \frac{1}{[(1 + \sigma^2(w)) + (z - \mu(w))^\top \Lambda(w)(z - \mu(w))]^{\sum_k \alpha_k}} \right]$$
(C.14)

$$= \mathbb{E}_{w \sim \text{Dir}(\alpha)} \left[ \frac{1}{(1 + \sigma^2(w))^{\sum_k \alpha_k}} \frac{1}{\left[ 1 + (z - \mu(w))^\top \frac{\Lambda(w)}{(1+\sigma^2(w))}(z - \mu(w)) \right]^{\sum_k \alpha_k}} \right]$$
(C.15)

$$= \mathbb{E}_{w \sim \text{Dir}(\alpha)} \left[ \frac{1}{(1 + \sigma^2(w))^{\sum_k \alpha_k}} \frac{1}{\left[ 1 + \frac{1}{\nu} \|z - \mu(w)\|^2_{\Omega(w)} \right]^{\sum_k \alpha_k}} \right].$$
(C.16)

The form in Eq. C.15 is useful because we can identify the quantity inside the integral as a function that is proportional to a multivariate $t$-distribution with location parameter $\mu(w)$, inverse scale matrix $\Omega(w) := \frac{\nu}{1+\sigma^2(w)}\Lambda(w)$, and degrees of freedom $\nu := 2\sum_k \alpha_k - D$.

Thus, we have shown that the Feynman parameterization allows us to to express the product of $t$-experts as a continuous mixture:

$$\prod_{k=1}^{K} q_k(z)^{\alpha_k} = \mathbb{E}_{w \sim \text{Dir}(\alpha)} \left[ (1 + \sigma^2(w))^{-\sum_k \alpha_k} \left[ 1 + \frac{1}{\nu} \|z - \mu(w)\|^2_{\Omega(w)} \right]^{-\sum_k \alpha_k} \right].$$
(C.17)

## C.2 Computing the normalizing constant (Result 2.2)

The definition of normalizing constant and Eq. C.17 gives

$$C_\alpha = \int \prod_{k=1}^{K} q_k(z)^{\alpha_k} dz = \int \mathbb{E}_{w \sim \text{Dir}(\alpha)} \left[ \frac{1}{(1 + \sigma^2(w))^{\sum_k \alpha_k}} \left[ 1 + \frac{1}{\nu} \|(z - \mu(w)\|^2_{\Omega(w)} \right]^{-\sum_k \alpha_k} \right] dz.$$
(C.18)

Because the integrand is non-negative, we can apply Fubini's (Tonelli's) theorem, to interchange the integral with the expectation:

$$C_\alpha = \mathbb{E}_{w \sim \text{Dir}(\alpha)} \left[ \frac{1}{(1 + \sigma^2(w))^{\sum_k \alpha_k}} \int \left[ 1 + \frac{1}{\nu} \|(z - \mu(w)\|^2_{\Omega(w)} \right]^{-\sum_k \alpha_k} dz \right].$$
(C.19)

Recall that the function in the inner integral is proportional to a multivariate $t$-distribution with location parameter $\mu(w)$, inverse scale matrix $\Omega(w) = \frac{\nu}{1+\sigma^2(w)}\Lambda(w)$, and degrees of freedom $\nu = 2\sum_k \alpha_k - D$. Thus, we can identify the integral as being equal to the inverse normalizing constant of this multivariate $t$-distribution multiplied by $(1 + \sigma^2(w))^{-\sum_k \alpha_k}$:

$$\frac{\pi^{\frac{D}{2}} \Gamma(\frac{\nu}{2}) \nu^{\frac{D}{2}} |\Omega(w)|^{-\frac{1}{2}}}{\Gamma(\frac{\nu+D}{2})} (1 + \sigma^2(w))^{-\sum_k \alpha_k} = \pi^{\frac{D}{2}} \frac{\Gamma(\sum_k \alpha_k - \frac{D}{2})}{\Gamma(\sum_k \alpha_k)} (1 + \sigma^2(w))^{\frac{D}{2} - \sum_k \alpha_k} |\Lambda(w)|^{-\frac{1}{2}},$$

where we simplified terms after expanding $\Omega(w)$. We also note that $\sum_k \alpha_k \text{sign}(|\Lambda_k|) > \frac{D}{2}$ is needed to ensure integrability.

After substituting the definition of $\nu$, the normalizing constant can be written as

$$C_\alpha = \int \prod_{k=1}^{K} q_k(z)^{\alpha_k} dz = \frac{\pi^{\frac{D}{2}} \Gamma(\frac{\nu}{2})}{\Gamma(\frac{\nu+D}{2})} \, \mathbb{E}_{w \sim \text{Dir}(\alpha)} \left[ |\Lambda(w)|^{-\frac{1}{2}} \left(1 + \sigma^2(w)\right)^{\frac{\nu}{2}} \right]. \tag{C.20}$$

Thus, this formulation of the normalizing constant turns the problem of an integral over $\mathbb{R}^D$ into an integral over the $K-1$ simplex.

# D  Score-based variational inference

In this section, we first describe a heuristic for placing the experts and then we perform an empirical study for this heuristic on an increasingly large number of experts. Then we perform an empirical study for the learning parameters of the variational inference algorithm.

## D.1  Heuristics for placing the experts

In this section, we provide one heuristic to address the main intuitive goal outlined in Section 3.1. Here, the goal is to select a large number of experts by first locating the mode(s) and then placing a large number of experts nearby to help shape the mode(s). The irrelevant experts are pruned away during the fitting the $\alpha$ weights via score matching.

We will now assume there is one mode, but this discussion can be extended to multiple modes. First, we locate the mode $z^*$ by hill climbing with the objective $\log p(z)$, where $p$ may be unnormalized. At the mode, we compute or estimate the curvature at $z^*$, which is given by $H(z^*) = \frac{1}{2} \nabla^2 \log p(z^*)$; this matrix arises from matching the second-order Taylor expansions of $\log p$ and $\log q_k$. The expert is placed at this mode

$$\mu_1 = z^*, \tag{D.1}$$
$$\Lambda_1 = -H(z^*). \tag{D.2}$$

Now we refine the mode by placing additional experts using the unnormalized target $\rho(z)$. Using a low discrepancy sequence, we generate $M$ candidate points from the hypercube

$$\left[ \mu_1 - s\sqrt{\text{diag}(\Lambda_1^{-1})}, \mu_1 + s\sqrt{\text{diag}(\Lambda_1^{-1})} \right]^D \tag{D.3}$$

where $s > 0$ is a scaling parameter. Then we resample the candidate points by sampling $N$ points with replacement according the weights: for a point $z_i$ in the candidate set, we compute

$$\omega_i \propto \exp(\beta \log \rho(z_i)) \tag{D.4}$$

where $\beta \in (0, 1]$ is a tempering parameter.

Finally, we refine the candidate set down to the final set of expert locations: beginning with the mode $\mu_1$, we greedily construct the set of experts by iterating through the candidate set and adding an element $\mu_k$ to the set of locations if $\|\mu_1 - \mu_k\| < \tau$, where $\tau > 0$ is a threshold parameter. The expert inverse scales are then set to $\Lambda_k = [-H(\mu_k)]_+$, where we project $[-H(\mu_k)]$ to the cone of positive semidefinite matrices.

This heuristic serves as a proof of concept and represents one possible way to place experts; future work will focus on more automated expert-selection strategies.

## D.2  Empirical study: expert-placing heuristics

In this section, we study the heuristic for algorithm placing on the sinh-arcsinh target described in Section 4 with $D = 5$, $\varepsilon_d = 0.3$, and $\delta_d = 0.7$. In all experiments, we used `tensorflow-probability` to generate a randomized Halton sequence, which was then transformed appropriately to obtain the desired hypercube.

We began by placing $M = 50{,}000$ candidate points, where we set $s = 15$, $\beta = 0.5$, $\tau = 6$. When choosing the locations, we greedily chose locations until hitting the desired number of experts

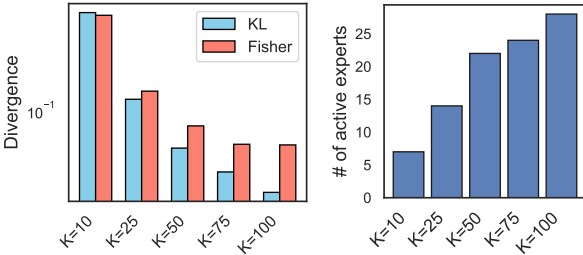

Figure D.1: Study of expert-placing heuristic. Left: KL and Fisher divergences for an increasing number of experts $K$. Right: The number of active experts after reweighting same sets of experts.

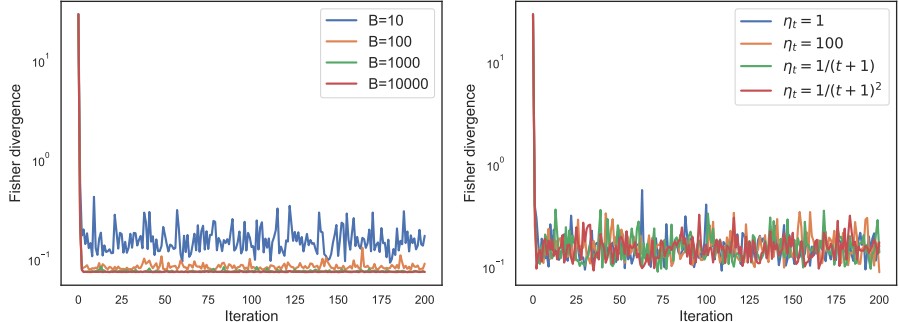

Figure D.2: Study of the learning parameters in score-based VI for the PoE family. Left: varying the number of samples $B$ while fixing the learning rate. Right: varying the learning rate schedule while fixing the batch size.

$K = 10, 25, 50, 75, 100$. We ran the variational inference algorithm for $T = 200$ iterations with $B = 20{,}000$ and a constant learning rate $\eta_t = 1$.

In Figure D.1, we report the metrics $\text{KL}(p; q)$ and $F(q; p)$. We observe that the KL and Fisher divergences decrease as the number of experts increase. We also report the number of active experts, i.e., the experts with non-zero weight. As the number of experts increases, the number of active experts increases slowly; in particular for $K = 50, 75, 100$, there were about 20-25 active experts. Thus, even with an overspecified number of experts, the method can learn a sparse set of experts.

**Computational cost of using more experts.** As we showed in this section, using more experts will result in an improved posterior accuracy. But it will also increase the computational cost of solving for the weights. Once the experts are fixed, determining the PoE parameters comes at a relatively small cost, since we need only solve a quadratic program to determine $\alpha$, which can be done in a matter of seconds, even in high dimensions. The computational cost of the constrained least squares solve is $O(K^3)$, which is not prohibitive for $K < 1000$. For the problems studied in this work, we did not require this many experts. For very large $K$, the cost will become prohibitive, and this issue provides motivation to work on better methods for expert selection in higher dimensions. In addition, the cost of computing scores is linear in the dimensionality of the latent variable $z$, and in many real-world problems, evaluating the unnormalizing target and its score is the dominating bottleneck.

### D.3 Empirical study: learning parameters

We continue studying the same target example as Appendix D.2. We used the same expert selection parameters, fixing $K = 50$. Now, we vary the learning parameters: the number of samples $B$, and the learning rate schedule $\eta_t$. For the number of samples, we considered $B = 10, 100, 1000, 10{,}000$; here we fixed $\eta_t = 1$. For the learning rate schedule, we fixed $B = 10$ and considered constant learning rates with $\eta_t = 1, 100$ and varying learning rates with $\eta_t = 1/(t + 1)$ and $\eta_t = 1/(t + 1)^2$. In all experiments, we initialized $\alpha^{(0)} = \mathbf{1}$. Here we observe that the larger batch sizes had faster and

more stable convergence. We additionally observe that the algorithm was not sensitive to the choice of $\eta_t$ considered.

# E   Proofs: convergence of score-based variational inference for the PoE family

## E.1   Statement of convergence theorem

For completeness and ease of reference, we restate the main definitions needed to state the theorem. In what follows, we assume that $\|\cdot\|$ denotes the L2 norm, and $\|\cdot\|_M$ denotes the Mahalanobis norm w.r.t. a matrix $M$.

The empirical Fisher divergence used in the objective is defined as

$$\widehat{\mathscr{D}}_t(\alpha) = \tfrac{1}{B}\sum_b \pi_b \big\| \nabla \log q(z_b|\alpha) - \nabla \log p(z_b) \big\|^2, \tag{E.1}$$

and is the first term in the variational inference objective function (Eq. 27). Let $\nabla \widehat{\mathscr{D}}_t$ and $\mathcal{H}[\widehat{\mathscr{D}}_t]$ denote the gradient and Hessian of this term with respect to the expert weights $\alpha$. With these definitions, we can state the following theorem.

---

**Theorem E.1.**   [Statement of Theorem 3.1] Suppose that $\mathscr{D}(q; p)$ in Eq. 1 is minimized by a unique $\alpha^* \in \mathscr{C}$, and also that for all $t \geq 0$ there exists some $\delta \geq 0$ such that $\mathbb{E}\big\| \tfrac{1}{2}\nabla \widehat{\mathscr{D}}_t(\alpha^*) \big\| \leq \delta$ and some $\lambda > 0$ such that $\mathcal{H}[\widehat{\mathscr{D}}_t] \succeq \lambda I$ almost surely. Then for constant learning rates $\eta_t \equiv \eta$, the expected error of the iterates satisfies

$$\mathbb{E}\big\| \alpha^{(t)} - \alpha^* \big\| \leq \left( \tfrac{1}{1+\eta\lambda} \right)^t \big\| \alpha^{(0)} - \alpha^* \big\| + \tfrac{\delta}{\lambda}. \tag{E.2}$$

---

## E.2   Proof of Theorem 3.1

*Proof.* The proof consists of several steps.

**Step 1.** *We begin by showing that minimizing the VI objective subject to the constraint $\alpha \in \mathscr{C}$ is equivalent to projecting an unconstrained solution into $\mathscr{C}$ with respect to a Mahalanobis norm.* To show this, we first recall that the objective can be written as

$$\widehat{\mathscr{D}}_t(\alpha) + \tfrac{1}{\eta_t}\|\alpha - \alpha^{(t)}\|^2 = \alpha^\top G_t \alpha + \alpha^\top h_t, \tag{E.3}$$

where

$$G_t := \tfrac{1}{B}\sum_{b=1}^B \pi_b Q_b^\top Q_b + \tfrac{1}{\eta_t}I, \qquad h_t := -2\left[ \tfrac{1}{B}\sum_{b=1}^B \pi_b Q_b^\top \nabla \log p(z_b) + \tfrac{1}{\eta_t}\alpha^{(t)} \right]. \tag{E.4}$$

Let $\tilde{\alpha}^{(t+1)}$ denote the *unconstrained minimizer* of this objective, which has the closed-form solution

$$\boxed{\tilde{\alpha}^{(t+1)} = -\tfrac{1}{2}G_t^{-1}h_t.} \tag{E.5}$$

Completing the square of the RHS of Eq. E.3, we have

$$\alpha^\top G_t \alpha + \alpha^\top h_t = (\alpha - \tilde{\alpha}^{(t+1)})^\top G_t (\alpha - \tilde{\alpha}^{(t+1)}) - \tfrac{1}{4}h_t^\top G_t^{-1}h_t. \tag{E.6}$$

Thus, we can we conclude that minimizing the VI objective is equivalent to projection onto $\mathscr{C}$ of the unconstrained solution in the Mahalanobis norm w.r.t. $G_t$:

$$\boxed{\arg\min_{\alpha \in \mathscr{C}} \widehat{\mathscr{D}}_t(\alpha) + \tfrac{1}{\eta_t}\|\alpha - \alpha^{(t)}\|^2 = \arg\min_{\alpha \in \mathscr{C}} \|\alpha - \tilde{\alpha}^{(t+1)}\|^2_{G_t}.} \tag{E.7}$$

**Step 2.** *Our next step is to show that bounds on the error of the unconstrained minimizer translate to bounds on the error of the minimizer in $\mathscr{C}$.* To this end, we define the error and unconstrained error, respectively, as

$$e_{t+1} := \alpha^{(t+1)} - \alpha^* \tag{E.8}$$

$$\tilde{e}_{t+1} := \tilde{\alpha}^{(t+1)} - \alpha^*. \tag{E.9}$$

Next we exploit the nonexpansive property of the projection in the Mahalanobis norm onto the closed convex set $\mathscr{C}$ [2, Proposition 4.8]. This property implies that the norm of the error $e_{t+1}$ is related to the weighted norm unconstrained error $\tilde{e}_{t+1}$. In particular, noting that $\alpha_{t+1}$ is the projection of $\tilde{\alpha}_{t+1}$, and $\alpha^*$ is the projection of $\alpha^*$ itself (since $\alpha^* \in \mathscr{C}$), we have that

$$\|e_{t+1}\|_{G_t}^2 = \|\alpha^{(t+1)} - \alpha^*\|_{G_t}^2 \leq \|\tilde{\alpha}^{(t+1)} - \alpha^*\|_{G_t}^2 = \|\tilde{e}_{t+1}\|_{G_t}^2 . \tag{E.10}$$

Thus, it suffices to bound the error of the unconstrained minimizer.

**Step 3.** *Our next step is to relate the unconstrained minimizer $\tilde{\alpha}^{(t+1)}$ to the gradient $\nabla\widehat{\mathscr{D}}_t$ at the optimizer $\alpha^*$ of $\mathscr{D}(q; p)$.* By doing so, we will make explicit the relationship between the unconstrained solution $\tilde{\alpha}^{(t+1)}$ and $\frac{1}{2}\nabla\widehat{\mathscr{D}}_t(\alpha^*)$, which provides a notion of misspecification in terms of the scores. Define the Hessian of $\widehat{\mathscr{D}}_t(\alpha)$ to be

$$H_t := \mathcal{H}[\widehat{\mathscr{D}}_t(\alpha)] = \nabla^2\widehat{\mathscr{D}}_t(\alpha) = \frac{1}{B}\sum_{b=1}^{B}\pi_b Q_b^\top Q_b. \tag{E.11}$$

Let $\delta_t$ denote one-half of the gradient of $\widehat{\mathscr{D}}_t$ at $\alpha^*$, which is given by

$$\delta_t := \tfrac{1}{2}\nabla\mathscr{D}_t(\alpha^*) = \tfrac{1}{2}\nabla\left(\frac{1}{B}\sum_{b=1}^{B}\pi_b\big\|\nabla\log q(z_b\,|\,\alpha) - \nabla\log p(z_b)\big\|_2^2\right)\bigg|_{\alpha=\alpha_*}$$

$$= \frac{1}{B}\sum_{b=1}^{B}\pi_b Q_b^\top\left(\nabla\log q(z_b\,|\,\alpha^*) - \nabla\log p(z_b)\right)$$

$$= \frac{1}{B}\sum_{b=1}^{B}\pi_b Q_b^\top\left(Q_b\alpha^* - \nabla\log p(z_b)\right). \tag{E.12}$$

Re-arranging Eq. E.12 and noting the definition of $H_t$ gives

$$\frac{1}{B}\sum_{b=1}^{B}\pi_b Q_b^\top\nabla\log p(z_b) = \frac{1}{B}\sum_{b=1}^{B}\pi_b Q_b^\top Q_b\alpha^* - \delta_t = H_t\alpha^* - \delta_t. \tag{E.13}$$

Using Eq. E.13 and the definition of $h_t$ in Eq. E.4, we can now re-express the unconstrained solution in Eq. E.5 as a function of $\delta_t$:

$$\tilde{\alpha}^{(t+1)} = -\tfrac{1}{2}G_t^{-1}\left[-2\left[\frac{1}{B}\sum_{b=1}^{B}\pi_b Q_b^\top\nabla\log p(z_b) + \tfrac{1}{\eta_t}\alpha^{(t)}\right]\right] \tag{E.14}$$

$$= G_t^{-1}\left[H_t\alpha^* + \tfrac{1}{\eta_t}\alpha^{(t)} - \delta_t\right]. \tag{E.15}$$

**Step 4.** *Our next step is to derive a recursion for the error in a Mahalanobis norm.* We start by analyzing the error of the unconstrained minimizer. It satisfies the following recursion w.r.t. $e_t$:

$$\tilde{e}_{t+1} = \tilde{\alpha}^{(t+1)} - \alpha^* \tag{E.16}$$

$$= G_t^{-1}\left[H_t\alpha^* + \tfrac{1}{\eta_t}\alpha^{(t)} - \delta_t\right] - \alpha^* \tag{E.17}$$

$$= G_t^{-1}[G_t - \tfrac{1}{\eta_t}I]\alpha^* + \tfrac{1}{\eta_t}G_t^{-1}\alpha^{(t)} - G_t^{-1}\delta_t - \alpha^* \tag{E.18}$$

$$= \alpha^* - \tfrac{1}{\eta_t}G_t^{-1}\alpha^* + \tfrac{1}{\eta_t}G_t^{-1}\alpha^{(t)} - G_t^{-1}\delta_t - \alpha^* \tag{E.19}$$

$$= \tfrac{1}{\eta_t}G_t^{-1}(\alpha^{(t)} - \alpha^*) - G_t^{-1}\delta_t \tag{E.20}$$

$$= G_t^{-1}(\tfrac{1}{\eta_t}e_t - \delta_t), \tag{E.21}$$

where in the third line we expanded the terms and substituted $H_t = G_t - \tfrac{1}{\eta_t}I$. Now we can translate this recursion on the unconstrained error into one for the actual error. To do so, we appeal to the

nonexpansive property of the projection onto $\mathscr{C}$. From Eq. E.21, we find in this way that

$$\|e_{t+1}\|_{G_t}^2 \leq \|\tilde{e}_{t+1}\|_{G_t}^2 \qquad \text{(nonexpansiveness)} \tag{E.22}$$

$$= \left\| G_t^{-1}\left( \tfrac{1}{\eta_t}e_t - \delta_t \right) \right\|_{G_t}^2 \qquad \text{(using Eq. E.21)} \tag{E.23}$$

$$= \left\| \tfrac{1}{\eta_t}e_t - \delta_t \right\|_{G_t^{-1}}^2. \tag{E.24}$$

**Step 5**. *Our next step is to translate the bounds on the error in the Mahalanobis norm into bounds on the error in the Euclidean norm.* In particular, the former can be bounded above and below by using the Euclidean norm and multiplying against eigenvalues of the matrices in the quadratic forms:

$$\lambda_{\min}(G_t)\,\|e_{t+1}\|_2^2 \leq \|e_{t+1}\|_{G_t}^2 \leq \left\| \tfrac{1}{\eta_t}e_t - \delta_t \right\|_{G_t^{-1}}^2 \leq \tfrac{1}{\lambda_{\min}(G_t)}\left\| \tfrac{1}{\eta_t}e_t - \delta_t \right\|_2^2. \tag{E.25}$$

Consequently, we obtain the following bound on the error $e_{t+1}$:

$$\|e_{t+1}\|_2^2 \leq \tfrac{1}{\lambda_{\min}(G_t)^2}\left\| \tfrac{1}{\eta_t}e_t - \delta_t \right\|_2^2 \tag{E.26}$$

$$= \tfrac{1}{\lambda_{\min}(G_t)^2}\left\| \tfrac{1}{\eta_t}(e_t - \eta_t\delta_t) \right\|_2^2 \tag{E.27}$$

$$= \tfrac{1}{\lambda_{\min}(G_t)^2}\tfrac{1}{\eta_t^2}\|e_t - \eta_t\delta_t\|_2^2. \tag{E.28}$$

Now, we note that $\lambda_{\min}(G_t) = \lambda_{\min}(H_t) + \tfrac{1}{\eta_t}$ and so $\eta_t\lambda_{\min}(G_t) = \eta_t\lambda_{\min}(H_t) + 1$. Making this substitution and taking square roots, we have

$$\|e_{t+1}\|_2 \leq \tfrac{1}{\eta_t\lambda_{\min}(G_t)}\|e_t - \eta_t\delta_t\|_2 \tag{E.29}$$

$$= \tfrac{1}{1+\eta_t\lambda_{\min}(H_t)}\|e_t - \eta_t\delta_t\|_2 \tag{E.30}$$

$$\leq \tfrac{1}{1+\eta_t\lambda}\|e_t - \eta_t\delta_t\|_2 \qquad \text{(using } \mathcal{H}[\widehat{\mathscr{D}}_t] \succeq \lambda I) \tag{E.31}$$

$$\leq \tfrac{1}{1+\eta_t\lambda}(\|e_t\|_2 + \|\eta_t\delta_t\|_2) \qquad \text{(triangle inequality)} \tag{E.32}$$

$$= \tfrac{1}{1+\eta_t\lambda}(\|e_t\|_2 + \eta_t\|\delta_t\|_2) \qquad \text{(using } \eta_t > 0) \tag{E.33}$$

**Step 6**. *Our next step is to translate this (per-iteration) recursion for the error in a particular run of the algorithm into bounds on the expected error after $t$ iterations.* Because each update of the iterates $\alpha^{(t+1)}$ depends on the samples $\{(z_b, \pi_b)\}_{b=1}^B \sim q(z \mid \alpha_t)$, the iterates are a stochastic process. When we take the expectation, we are taking expectation with respect to the filtration defined by $\{\alpha^{(t)}\}_{t\geq0}$.

We proceed by taking expectations on both sides of the recursion in Eq. E.33 and using our assumption that $\mathbb{E}\big\|\tfrac{1}{2}\nabla\widehat{\mathscr{D}}_t(\alpha^*)\big\| \leq \delta$. In this way we find

$$\mathbb{E}\,\|e_{t+1}\|_2 \leq \tfrac{1}{1+\eta_t\lambda}\mathbb{E}\|e_t\|_2 + \tfrac{\eta_t\delta}{1+\eta_t\lambda}. \tag{E.34}$$

Unrolling this recurrence, we obtain the bound

$$\mathbb{E}\,\|e_t\|_2 \leq \rho_t\,\|e_0\|_2 + \nu_t, \tag{E.35}$$

where

$$\rho_t := \prod_{j=1}^{t-1}\tfrac{1}{1+\eta_j\lambda}, \tag{E.36}$$

$$\nu_t := \sum_{i=0}^{t-1}\left( \prod_{j=i+1}^{t-1}\tfrac{1}{1+\eta_j\lambda} \right)\tfrac{\eta_i\delta}{1+\eta_i\lambda}. \tag{E.37}$$

**Step 7**. *Our final step is to examine the above bound for a constant step size $\eta_t \equiv \eta$.* Define the quantity

$$\varphi(\eta, \lambda) := \tfrac{1}{1+\eta\lambda} < 1. \tag{E.38}$$

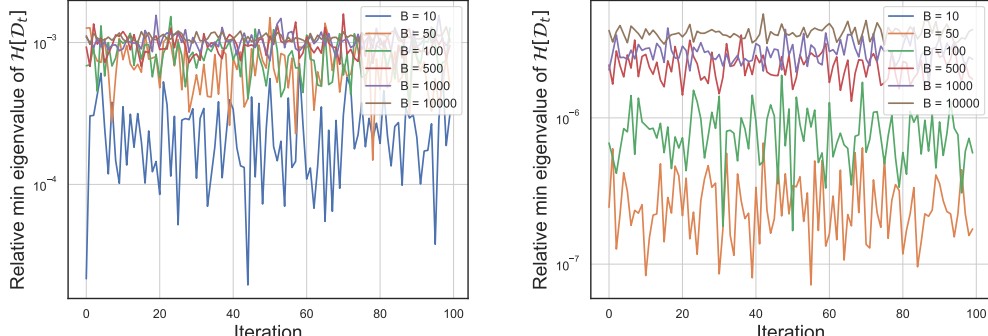

Figure E.1: Relative minimum eigenvalues with $K = 20$ (left) and $K = 100$ (right) experts.

In this case we have that

$$\rho_t = \varphi(\eta, \lambda)^t \tag{E.39}$$

$$\nu_t = \sum_{i=0}^{t-1} \varphi(\eta, \lambda)^{t-i} \eta \delta \tag{E.40}$$

$$= \eta \delta \sum_{j=1}^{t} \varphi(\eta, \lambda)^j \qquad \text{(change of variables } j = t - i) \tag{E.41}$$

$$= \eta \delta \frac{\varphi(\eta, \lambda) - \varphi(\eta, \lambda)^{t+1}}{1 - \varphi(\eta, \lambda)} \qquad \text{(sum of geometric series)} \tag{E.42}$$

$$= \eta \delta \frac{1+\eta\lambda}{\eta\lambda} \left[ \varphi(\eta, \lambda) - \varphi(\eta, \lambda)^{t+1} \right] \tag{E.43}$$

$$= \frac{\delta}{\lambda} \left[ 1 - \varphi(\eta, \lambda)^t \right] \tag{E.44}$$

$$= \frac{\delta}{\lambda} [1 - \rho_t] \tag{E.45}$$

$$\leq \frac{\delta}{\lambda}, \tag{E.46}$$

and the theorem is proved by substituting the results in Eq. E.39 and Eq. E.46 into Eq. E.35. $\qquad \square$

### E.3 Empirical verification of strong convexity constant $\lambda$

We ran 100 iterations of Algorithm 1 on the sinh-arcsinh target described in Section 4 with $D = 5$, $\varepsilon_d = 0.3$, and $\delta_d = 0.7$. We ran the algorithm for an increasing number of samples $B$. In all iterations $\lambda > 0$. For each iteration, we computed the relative minimum eigenvalue of $\mathcal{H}[\widehat{D}_t]$:

$$r_{\min} = \lambda_{\min}/\lambda_{\max}, \tag{E.47}$$

where higher values indicate a more well-behaved positive definite matrix. In Figure E.1, we plot the relative eigenvalues for $K = 20$ and $K = 100$. In both cases, we found that larger batch sizes tended to lead to more well-behaved matrices. For $K = 100$ and $B = 10$, the matrix was effectively singular, and so we omit that case from the plot.

### E.4 Relationship with broader literature

Finally, we highlight the relationship of this convergence result with the broader convergence literature. Theorem 3.1 shares similarities with the convergence behavior of stochastic gradient descent (SGD) applied to strongly convex functions, as established in prior works [13, 39, 40]. However, traditional SGD analyses typically assume a fixed objective function throughout the optimization process. In contrast, our setting involves an objective function that changes at each iteration, rendering standard SGD theory inapplicable.

The online learning framework accommodates such scenarios by allowing the objective function to vary over time, even in an adversarial manner [41]. Within this framework, convergence results often focus on bounding the *regret*, defined as the cumulative difference between the loss incurred by the algorithm at each iteration and the loss of a comparator chosen in hindsight. Notably, even in

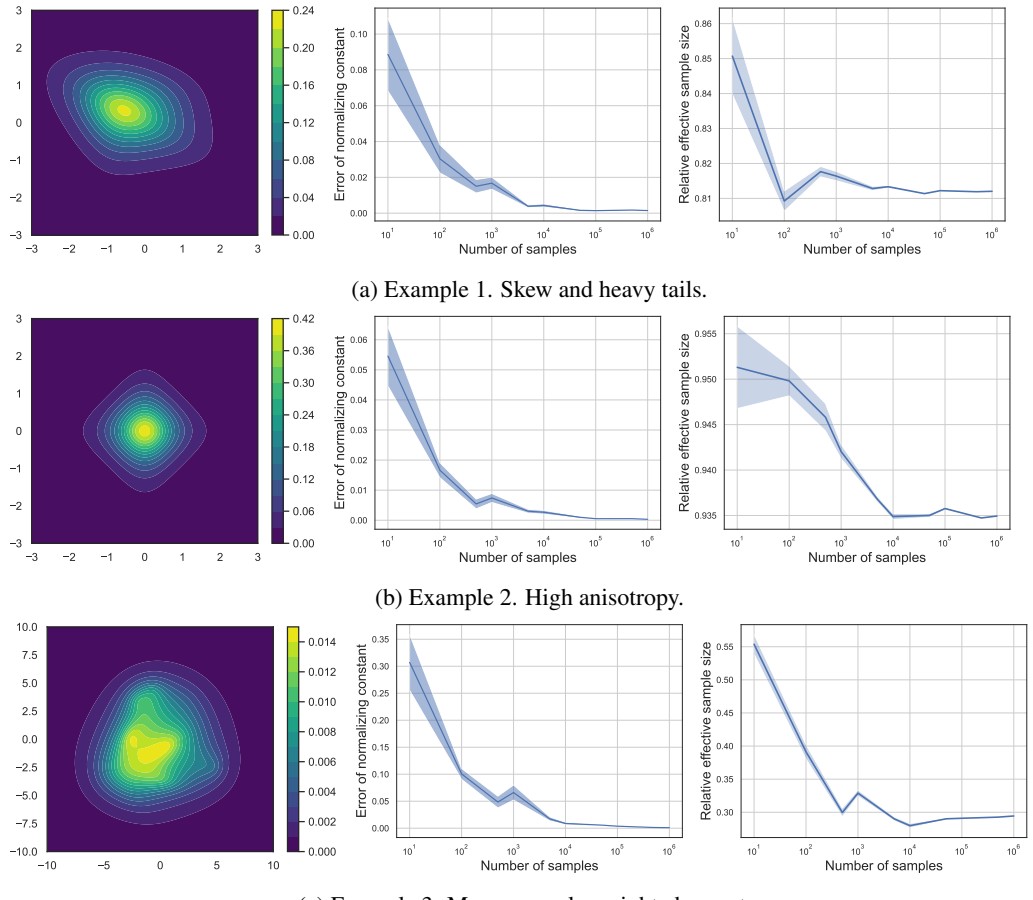

(a) Example 1. Skew and heavy tails.

(b) Example 2. High anisotropy.

(c) Example 3. Many sparsely weighted experts.

Figure F.1: Empirical study for the normalizing constant estimate (second column) and the sampling procedure (third column) for several PoE densities (first column). For each metric, we compute the mean value with respect to 10 random seeds. The shaded region represents the standard error taken with respect to 10 random seeds.

the strongly convex setting, these results provide bounds on regret rather than on the distance to the optimal solution.

In contrast, Theorem 3.1 establishes a bound on the expected $\ell_2$ distance between the iterates and the optimal solution $\alpha^*$, offering a different perspective on convergence in the presence of time-varying objective functions.

# F    Experiments: additional studies

In Appendix D.2 and Appendix D.3, we provided empirical studies regarding the expert selection heuristic and the VI algorithm to reweight the experts, and in Appendix E.3 we provided an empirical study to supplement the theoretical convergence results.

In this section, we begin by providing an empirical study on the normalizing constant and sampling procedure. In addition, we expand on the details and also provide additional results related to the evaluation of the score-based VI method for the PoE family.

## F.1 Empirical study: normalizing constant and sampling procedure

In this section, we perform a study for the estimate of the normalizing constant using the Feynman parameterization. We also study the performance of the importance sampling procedure. We considered the following three examples.

**Example 1: skew and heavy tails.** Consider a product of 3 experts in $D = 2$ with locations

$$\mu_1 = [-1, -1]^\top, \mu_2 = [0, 0]^\top, \mu_3 = [1, 1]^\top,$$

inverse scale matrices

$$\Lambda_1 = \begin{bmatrix} 1 & 0 \\ 0 & 1/3 \end{bmatrix}, \Lambda_2 = \begin{bmatrix} 1/3 & 1/2 \\ 1/2 & 1 \end{bmatrix}, \Lambda_3 = \begin{bmatrix} 1/3 & 0 \\ 0 & 1 \end{bmatrix},$$

and weights $\alpha = [1, 1.2, 1]^\top$.

**Example 2: high anisotropy.** In our second example, we place 2 experts with locations

$$\mu_1 = [0, 0]^\top, \mu_2 = [0, 0]^\top,$$

inverse scale matrices

$$\Lambda_1 = \begin{bmatrix} 1 & 0 \\ 0 & 1/500 \end{bmatrix}, \Lambda_2 = \begin{bmatrix} 1/500 & 0 \\ 0 & 1 \end{bmatrix},$$

and weights $\alpha = [2.0, 2.0]^\top$.

**Example 3: many sparsely weighted experts.** Here we considered $K = 100$ experts, but with the majority of $\alpha$ weights on 30 "active" experts. We generated the expert locations $\mu_k \sim \text{unif}[-5, 5]$. For the expert inverse scale matrices, we randomly generated two eigenvalues $\lambda_k$ uniformly from $[0.05, 2]$, and a random rotation $R$, and then formed the matrix $\Lambda_k = R \, \text{diag}(\lambda_k) R^\top$. The weight for the $\alpha$ vector was constructed so that the first 30 received weight 0.25, and the remaining received weight $10^{-12}$; then the entries were randomly shuffled.

For all three of the above examples, we computed a Monte Carlo estimate of the normalizing constant:

$$C_\alpha \approx \frac{\pi^{\frac{D}{2}} \Gamma(\frac{\nu}{2})}{\Gamma(\frac{\nu+D}{2})} \sum_{b=1}^{B} \left[ |\Lambda(w_b)|^{-\frac{1}{2}} (1 + \sigma^2(w_b))^{\frac{\nu}{2}} \right], \qquad w_b \stackrel{\text{iid}}{\sim} \text{Dir}(\alpha). \tag{F.1}$$

As "ground truth," we computed the estimate of the normalizing constant using 5 million samples. Then, we estimated the normalizing constant using an increasing number of samples $B = 10, 100, 500, 1000, 5000, 10,000, 50,000, 100,000, 500,000, 1,000,000$. In Figure F.1, we plot the mean absolute error between the $B$ samples and the "true" value, where the mean is taken over 10 random seeds.

For the sampling study, we ran the importance sampling procedure for the same increasing numbers of samples $B$ as above. The sampling procedure outputs normalized weights $\{\pi_b\}_{b=1}^{B}$. Using these weights, we then calculated (an estimate of) the effective sample size (ESS) [34], which provides a measure of how much efficiency in sampling is lost to weight variability:

$$\widehat{\text{ESS}} = \frac{(\sum_{b=1}^{B} \pi_b)^2}{\sum_{b=1}^{B} \pi_b^2}. \tag{F.2}$$

In Figure F.1, we plot the relative ESS, which divides the ESS by $B$. The relative ESS measures the efficiency of the sampling procedure, where values closer to 1 are better.

Here we find that in the first two examples, the sampling procedure has high relative ESS, with values above 0.8. For the third example, there is a decrease in efficiency, with values roughly between 0.3–0.5. While this procedure can still be effective for variational inference, it indicates that more samples may need to be taken.

The final result also suggests several opportunities for improvement. First, the sampler could be specialized to take advantage of the sparsity of the $\alpha$ weights. These results also suggest that a more compact set of experts may lead to more effective variational inference procedures. This motivates developing more advanced approaches for expert selection than the heuristic considered here, which places down a large number of experts and then reweights their importance. However, as we observe in the later VI experiments, given enough samples, the VI fits are still able to provide accurate posterior approximations, even when using a large number of experts.

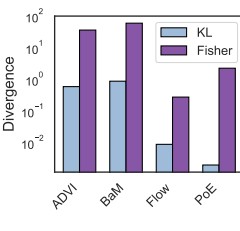
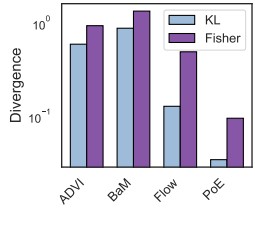
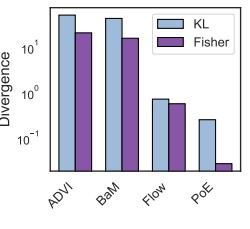

(a) Rosenbrock, $D = 2$        (b) Sinh-arcsinh $D = 5$        (c) Sinh-arcsinh $D = 10$

Figure F.2: Additional synthetic target examples

## F.2    Target distributions in Section 4

**1) Gaussian mixture target:** We constructed a mixture of 3 Gaussian distributions with weights $0.3, 0.4, 0.3$. The means were set to $\mu_1 = [-1.0, 0.0]^\top, \mu_2 = [1.0, 0.0]^\top, \mu_3 = [0.0, 1.0]^\top$, and the covariances were set to $\Sigma_1 = \Sigma_2 = \left[ \begin{smallmatrix} 0.5 & 0.0 \\ 0.0 & 0.5, \end{smallmatrix} \right]$ and $\Sigma_3 = \left[ \begin{smallmatrix} 1.0 & 0.5 \\ 0.5 & 1.0, \end{smallmatrix} \right]$.

**2) PoE target:** We constructed a multimodal PoE target with locations $\mu_1 = [-2, -2]^\top$ and $\mu_2 = [2, 2]^\top$. The inverse scale matrices were set to $\Lambda_1 = \left[ \begin{smallmatrix} 1.0 & 0.5 \\ 0.2 & 1.0, \end{smallmatrix} \right]$ and $\Lambda_2 = \left[ \begin{smallmatrix} 1.0 & 0.1 \\ 0.1 & 1.0, \end{smallmatrix} \right]$. The $\alpha$ weights were all set to 1.

**3) Diamond target:** We constructed a PoE target in $D=2$ with $\mu_1 = \mu_2 = [0,0]^\top$, $\Lambda_1 = \left[ \begin{smallmatrix} 100 & 0 \\ 0 & 1 \end{smallmatrix} \right]^{-1}$, $\Lambda_2 = \left[ \begin{smallmatrix} 1 & 0 \\ 0 & 100 \end{smallmatrix} \right]^{-1}$, and $\alpha = [1.2, 1.2]^\top$. This target density forms a diamond-like shape.

**4) Funnel target:** The model is $z_1 \sim \mathcal{N}(0, \sigma^2), z_2, \ldots, z_D \sim \mathcal{N}(0, \exp(z_1/2))$, where we set $\sigma^2 = 1.1$.

## F.3    Additional synthetic target results

Additional synthetic target results are in Figure F.2. The Rosenbrock target used was

$$\log p(z) = -[(1 - z_1)^2 + 2(z_2 - z_1^2)^2]. \tag{F.3}$$

The normalizing constant was estimated via quadrature and samples from $p$ were generated via a blackjax [5] implementation of HMC. The sinh-arcsinh targets used were all positively skewed ($\varepsilon = 0.3$) and heavy-tailed ($\tau_d = 0.7$).

## F.4    Experimental Setup for Section 4

**Computing resources.**    All experiments were run on CPU. We used a Linux workstation with a 32-core Intel processor and with 503 GB of memory.

**Tuning parameters for baseline methods.**    The normalizing flow variational family was based on a realNVP [10]. We followed the implementation details of the package `FlowMC` [55]. The base distribution was a standard multivariate normal. In all lower-dimensional synthetic experiments ($D < 50$), we fixed the number of layers to 8 and the number of hidden units to 128. For the 50-dimensional sinh-arcsinh target and the posterior-db experiments, we set the number of layers to 16 and the number of hidden units to 128. The flow weights in every coupling layer were initialized randomly using a draw from normal centered at 0 with scale=1e-4.

For the Gaussian variational families, the initial values were by default set to a standard Gaussian. For the stochastic gradient-based methods (ADVI, Flow), we optimized the ELBO using Adam [25]. In order to choose the learning rate, we conducted pilot studies where we ran each method for at least 2000 iterations, using a grid of learning rate values $[0.001, 0.005, 0.01, 0.02, 0.03]$. The learning rate chosen was the one resulting in the lowest loss value. Some examples are shown in Figure F.3. For BaM, the learning rate schedule was fixed to $\frac{BD}{(1+t)}$ in all experiments; due to the highly non-Gaussian target problems considered, the performance of this method could likely be improved with additional tuning. For the lower-dimensional targets ($D < 50$), all three methods used the same fixed batch size $B = 64$ and number of iterations $T = 5000$. For the 50-dimensional target, all three methods used a batch size of $B = 128$ and number of iterations $T = 10,000$. Thus, we allocated the same budget

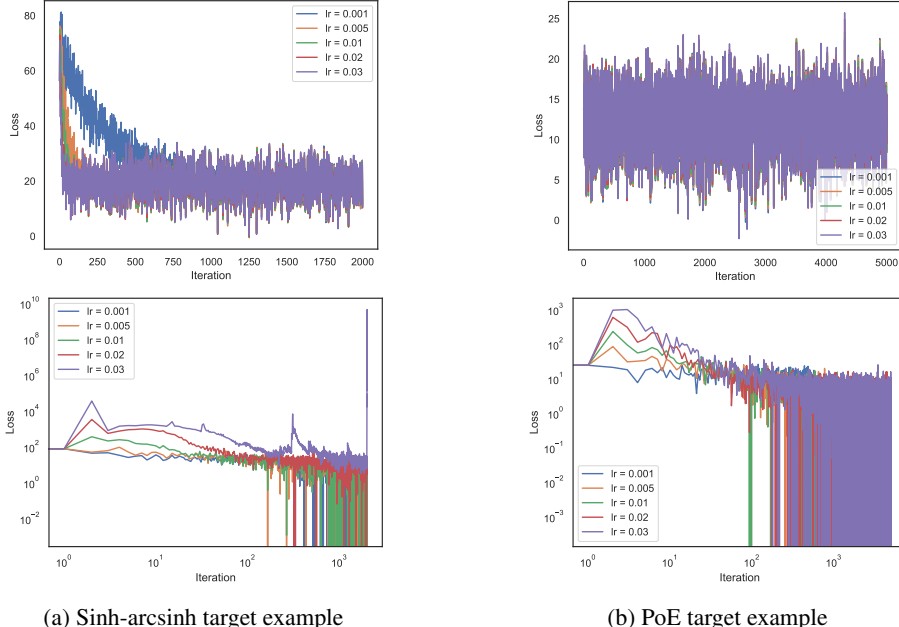

(a) Sinh-arcsinh target example        (b) PoE target example

Figure F.3: Examples of pilot studies conducted for hyperparameter grid search. Top: ADVI. Bottom: Normalizing flow.

Table F.1: Summary of PoE settings for synthetic targets.

| **Target** | $K$ | Active | $M$ | $s$ | $\beta$ |
|---|---|---|---|---|---|
| Gaussian mixture | 143 | 93 | 50,000 | 28 | 0.4 |
| Product of experts | 157 | 22 | 50,000 | 28 | 0.5 |
| Diamond | 60 | 47 | 50,000 | 28 | 0.1 |
| Funnel | 50 | 17 | 50,000 | 15 | 0.5 |
| Sinh-arcsinh ($D$=5) | 100 | 27 | 50,000 | 15 | 0.5 |
| Sinh-arcsinh ($D$=10) | 480 | 243 | 1,000,000 | 28 | 0.5 |
| Sinh-arcsinh ($D$=50) | 400 | 184 | 3,500,000 | 28 | 0.4 |
| Rosenbrock | 90 | 26 | 3,500,000 | 20 | 0.5 |

in terms of number of gradient evaluations of $\nabla \log p$ (we note that in many real-world problems, evaluating the score of the target dominates the computational cost).

**Details on PoE VI procedure.** The PoE-VI code was implemented using Python. To solve the quadratic program in Algorithm 1, we used the `qpsolvers` package with the solver `jaxopt_osqp`, which is a JAX wrapper around the OSQP solver [50]. In order to enforce strict inequality constraints for integrability, we added a small slack parameter $\varepsilon = $ 1e-12. We note that when not all inverse scale matrices are chosen to be positive definite, a sufficient condition for integrability is $2 \sum_k \alpha_k \, \text{sign}(|\Lambda_k|) > D$, but we found that it was often overly strong. Instead we used the constraint $2 \sum_k \alpha_k > D$ and verified that the solution resulted in an integrable density. By default, we used a batch size of $B = 10,000$ and a constant learning rate of 100; the effects of varying these parameters were investigated in earlier experiments. For all experiments, we used the default sampling procedure described in Eq. 25. For the normalizing constant, 500,000 samples were used to estimate the normalizing constant, which was used only when computing the estimate of the forward KL divergence. Based on the learning results in Appendix D.2, we set the learning rate to $\lambda_t = 1$, and we ran the algorithm for $T = 20$ iterations. In all experiments, we initialized $\alpha^{(0)} = \mathbf{1}$.

In Table F.1, we summarize the expert selection details used on each of the synthetic targets, along with the number of experts selected and number of active experts in the final PoE.

**posterior-db targets.** For all posterior-db experiments, we use the BridgeStan library [44] to first transform each target distribution to have support in $\mathbb{R}^D$ before fitting the VI procedures.

**posterior-db: garch11.** This model is a discrete-time volatility model with a given constant $\sigma_1$. The model uses improper uniform priors on the parameters $\mu$ and $\alpha_0 > 0$, resulting in:

$$\mu \sim \text{uniform} \tag{F.4}$$

$$\alpha_0 \sim \text{half-uniform} \tag{F.5}$$

$$\alpha_1 \sim \text{uniform}(0, 1) \tag{F.6}$$

$$\beta_1 \sim \text{uniform}(0, 1 - \alpha_1). \tag{F.7}$$

For each time step $t = 2, \ldots, \tilde{T}$:

$$\sigma_t^2 = \alpha_0 + \alpha_1(y_{t-1} - \mu)^2 + \beta_1 \sigma_{t-1}^2 \tag{F.8}$$

$$\varepsilon_t \sim \mathcal{N}(0, 1) \tag{F.9}$$

$$y_t = \mu + \sigma_t + \varepsilon_t. \tag{F.10}$$

The PoE method used $K = 100$ experts, with 24 experts active in the final variational approximation. Experts were chosen with $M = 80{,}000$, $s = 12$, and $\beta = 0.4$.

**posterior-db: eight-schools.** This is a classic hierarchical model where each school $j$ has its own effect $\theta_j$. In this model, the global hyperpriors are

$$\mu \sim \mathcal{N}(0, 5^2) \tag{F.11}$$

$$\tau \sim \text{half-Cauchy}(0, 5) \tag{F.12}$$

For each school $j = 1, \ldots, 8$, the parameters and observations are generated as:

$$\eta_j \sim \mathcal{N}(0, 1) \tag{F.13}$$

$$\theta_j \sim \mu + \tau\eta_j \tag{F.14}$$

$$y_j \sim \mathcal{N}(\theta_j, \sigma_j^2) \tag{F.15}$$

The PoE method used $K = 50$ experts, with 24 experts active in the final variational approximation. Experts were chosen with $M = 300{,}000$, $s = 14$, and $\beta = 0.45$.

**posterior-db: gp-regr.** This example is a Gaussian process regression model with mean 0 and covariance function

$$K(x_i, x_j) = \alpha^2 \exp\left(-\frac{(x_i - x_j)^2}{2\rho^2}\right). \tag{F.16}$$

Given $N$ data points, let $\tilde{K}$ denote the $N \times N$ gram matrix. The generative model is:

$$\rho \sim \text{gamma}(25, 4) \tag{F.17}$$

$$\alpha \sim \text{truncated-normal}(0, 2^2) \tag{F.18}$$

$$\sigma \sim \text{truncated-normal}(0, 1) \tag{F.19}$$

$$f \sim \mathcal{N}(0, \tilde{K}) \tag{F.20}$$

$$y \sim \mathcal{N}(f, \sigma^2 I). \tag{F.21}$$

The PoE method used $K = 70$ experts, with 53 experts active in the final variational approximation. Experts were chosen with $M = 10{,}000$, $s = 10$, and $\beta = 0.6$.

