# OpenReview forum: "Fisher meets Feynman: score-based variational inference with a product of experts"
_NeurIPS.cc/2025/Conference — NeurIPS 2025 spotlight_

### Official Review · Reviewer_UfQj · 2025-06-27

**Clarity:** 3
**Significance:** 3
**Originality:** 3
**Rating:** 5
**Confidence:** 3

**Summary:**

This paper addresses the variational inference (VI) where the approximate variational distribution is assumed to be a weighted product of experts, and each expert is a multivariate t-distribution. To optimize the expert weights, this paper first use Feynman identity to reformulate the product of t-distributions as a latent variable model where the latent variable (denoted as w) lies in the simplex and the observed variable (denoted as z) is conditionally t-distributed given w. Then propose to minimize the Fisher divergence between the PoE and the target density. Some experiments on synthetic and real-world datasets are conducted to demonstrate the performance of the proposed method.

**Questions:**

See the weaknesses.

**Ethical Concerns:**

["NO or VERY MINOR ethics concerns only"]

**Final Justification:**

Considering rebuttal and discussions with authors, I am happy to increase my score.

**Limitations:**

Yes

**Paper Formatting Concerns:**

I do not notice any major formatting issues in this paper

**Quality:**

3

**Strengths And Weaknesses:**

Strengths:
Overall, I enjoy reading this paper.
1. The idea of using the Feynman identity to reformulate the PoE into a continuous mixture of t-distributions is interesting. In this formulation, the mixture weights w are Dirichlet-dsitributed, and the variable z follows t-distribution given w. This makes sampling from the PoE more tractable: one can sample w from a Dirichlet and then sample z from conditional t-distribution.

2. The Fisher div between the PoE and target is optimized using "batch-and-match" strategy. Since this requires sampling from PoE for each iteration, the Feynman identity-based reformulation provides a practical advantages by facilitating this sampling process.

3. Experimental results demonstrate that the proposed PoE model captures the heavy tails of the target distributions more effectively than Gaussian approximations, benefiting from the properties of t-distributions.

Weaknesses:
1. The proposed algorithm only learns the expert weights (\alpha ), while the set of expert distributions (t-distributions) must be predefined. The parameters of these experts remain fixed during training, which could limit the model’s ability to approximate complex target distributions.

2. The Feynman identity-based reformulation uses the fact that the density of t-distribution is proportional to the inverse of a quadratic from. This property is essential for applying  The Feynman identity, and may not generalize easily to other distribution families. This might limit the broader applicability of the proposed approach.

---

> ### Author Rebuttal · Authors · 2025-07-31
>
> Thank you for your time in reading our paper. We are happy to hear that you enjoyed reading this paper and appreciated the novelty of the Feynman identity approach and the tractability it adds to this PoE framework.
>
> We respond to your comments and feedback below.
>
> **1. Fixed experts and flexibility**
>
> > “The proposed algorithm only learns the expert weights (\alpha ), while the set of expert distributions (t-distributions) must be predefined. The parameters of these experts remain fixed during training, which could limit the model’s ability to approximate complex target distributions.”
>
> There are many types of approaches to forming a flexible density estimator. One way is the “basis set” view, where you attempt to learn the coefficients of a basis expansion. We deliberately adopt a view that is reminiscent of the basis-expansion perspective, where we begin in a large, expressive set of experts that is then reweighted by minimizing the score divergence. Indeed, Figure 1 (leftmost and rightmost panels) shows that with just two fixed experts, changing the weights can result in very different shaped densities.  Furthermore, we show in our experiments that the variational approximation can approximate a variety of target distributions, including challenging geometries such as banana, funnel, cross, and multimodal target distributions (see also new results added, see our response to Review TPhU).
>
> Finally, we note that placing experts randomly and reweighting is one of the simplest first approaches to optimizing this family; there are other more sophisticated approaches that we are actively investigating for future work (see our response to Review xP2U).
>
> **2. Feynman identity and broader applicability**
> > “The Feynman identity-based reformulation uses the fact that the density of t-distribution is proportional to the inverse of a quadratic from. This property is essential for applying The Feynman identity, and may not generalize easily to other distribution families. This might limit the broader applicability of the proposed approach.”
>
> We note that the Feynman representation of the product of t-experts leads to an already fairly rich variational family: we can model distributions with different tail weights, and we note that the experts themselves don’t need to be normalized. It is an interesting question what other family of functions remain “closed” under convex combinations, are easy to integrate, and lead to an expressive family of densities. We chose to focus on the t-distribution case given its use in the literature [1], and were able to show in Result 2.1 that this already produces an expressive variational family.
>
> [1] Welling et al. Learning sparse topographic representations with products of student-t distributions. NeurIPS, 2002.

---

> > ### Comment · Reviewer_UfQj · 2025-08-04
> > **I have updated my score**
> >
> > Thank the authors for informative rebuttal. It clarified my concerns. I am happy to increase my score to "Accept".

---

### Official Review · Reviewer_TPhU · 2025-07-01

**Clarity:** 4
**Significance:** 2
**Originality:** 4
**Rating:** 5
**Confidence:** 4

**Summary:**

This work presents a novel score-based BBVI procedure based on a Product of Experts (PoE) that models skewed and/or heavy tailed distributions more accurately than state-of-the-art methods.

To that end, the authors introduce a new family of PoEs, each proportional to a multivariate t-distribution. Notably, they show how to draw samples from this (complex) family and how to estimate its normalizing constant, by transforming the problem via the Feynman identity (Section 2).

With this PoE (plus importance sampling tricks), the authors derive an iterative score-matching procedure to optimize the PoE's geometric weighting of members, via minimization of a regularized Fisher divergence (Section 3). This minimization is proven to be a convex quadratic problem, hence resulting in rates of convergence to the neighborhood of an optimal weighting of experts.

The empirical experiments in synthetic and a Bayesian posterior benchmark of Section 4 illustrate how BBVI with this PoE better approximates skewed and/or heavy tailed densities.

**Questions:**

- Can the authors elaborate on the benefits (and if any, drawbacks) of the proposed PoE based BBVI technique beyond the skewed and heavy-tailed distributions presented in Section 4?
    - Namely, what is the performance of PoE-BBVI in other posteriors?
    - Is it better/comparable/worse than alternatives?

- Can the authors elaborate on the computational cost/complexity of computing the PoE within the BBVI framework?
    - what is the tradeoff between compute time and posterior accuracy?
    - can these metrics be reported for the evaluations in Section 4?

- Even though it is listed as a limitation, more details on the initialization procedure would be appreciated:
    - How sensitive are the presented results to a good initialization of the fixed collection of experts selected upfront?
    - What is the number of experts used in the experiments, and how do results change with respect to varying it?
    - What's the computational complexity of such initialization procedure?

**Ethical Concerns:**

["NO or VERY MINOR ethics concerns only"]

**Final Justification:**

I have increased my score towards acceptance, as the authors' rebuttal has been thorough and informative, clarifying many of the minor concerns raised, as well as by providing new empirical evidence of the benefits of the proposed method.

**Limitations:**

Yes

**Quality:**

3

**Strengths And Weaknesses:**

**The contributions of this work are two-fold**:
- The use of the Feynman identity to demonstrate how to draw samples, compute the marginal and estimate quantities of interest from a PoE with experts proportional to multivariate t-distributions (efficiently over the simplex, instead of over the reals)
- The use of such PoE within BBVI to derive an iterative score-matching procedure for optimization of the geometric weighting of experts for successful approximation to skewed and heavy-tailed distributions

Additionally, I would highlight the theoretical analysis of the weighting of experts procedure, resulting in convergence guarantees.

As the **main weaknesses** of this work, I would identify:
- The heuristic proposed to select the experts (and the lack of evaluation of its computational cost and impact on performance)
- The (relatively) complex procedure to attain gains (only?) on skewed and heavy-tailed posteriors
    - See my questions below on the method's tradeoffs, and whether results in other posteriors are comparable to SOTA

---

> ### Author Rebuttal · Authors · 2025-07-31
>
> We appreciate your feedback and engagement with the paper. We are glad to hear you found the paper clearly written and highly original. Thanks also for highlighting our _theoretical analysis of the algorithm_ – due to the changing objective function, we could not apply traditional SGD theory for strongly convex functions, and needed to develop a specialized analysis of the algorithm.
>
> Below, we answer your questions and comment on feedback in the review.
>
> **1. Performance beyond skewed / heavy-tailed posteriors**
>
> > “Can the authors elaborate on the benefits (and if any, drawbacks) of the proposed PoE based BBVI technique beyond the skewed and heavy-tailed distributions presented in Section 4?
> Namely, what is the performance of PoE-BBVI in other posteriors?
> Is it better/comparable/worse than alternatives?”
>
> First, we would like to emphasize that we evaluated more than _just_ skewed and heavy-tailed posteriors – these posteriors were emphasized because some standard baselines do not work well in these settings (e.g., Gaussians cannot model skew or heavy tails, and the standard flow-based models are not designed for heavy tails).
>
> In our experiments, not only did we study skew and heavy tails, we also studied examples that are _light tailed_ (e.g., garch, funnel, 8-schools, gp-regr) and have _complex geometries_ (e.g., the funnel, the cross, and 8-schools). Furthermore, we found that our PoE method achieved the best resulting Fisher divergence on the posteriordb targets in Figure 4.
>
> That said, we appreciate your curiosity in understanding the performance in a broader class of models. We have added a few other examples that showcase different target distributions. The _summary of target distributions_ is below, where we have indicated which ones are new:
> 1. Cross shape
> 2. Funnel
> 3. Skewed and heavy-tailed
> 4. Banana / Rosenbrock **(new)**
> 5. Multimodal, well-separated modes **(new)**
> 6. Multimodal, modes from mixing distributions **(new)**
> 7. PosteriorDB - garch11: heavy-tailed + skewed
> 8. PosteriorDB - 8schools: funnel
> 9. PosteriorDB - gp-regr: light tailed and symmetric
> 10. PosteriorDB - sesame: **(new)**
>
> We believe this provides a diverse set of target distributions that often arise in hierarchical modeling settings. For the final revision, we plan to include additional posteriorDB models.
>
> _**Additional results**_
>
> **Rosenbrock function (banana)**
>
> | Approach |   KL  | Fisher |
> |----------|-------|--------|
> | BaM      | 0.89  | 60.66  |
> | ADVI     | 0.60  | 36.71  |
> | Flow     | 0.009 | 0.28  |
> | PoE      | 0.002 | 2.29  |
>
>
>
> **Multimodal example (well-separated modes)**
>
> | Approach |  KL  | Fisher |
> |----------|------|--------|
> | BaM      | 1.59 |  1.14  |
> | ADVI     | 0.93 |  0.64  |
> | Flow     | 0.05 |  0.12  |
> | PoE      | 0.08 |  0.07  |
>
>
> **Multimodal example (modes from mixing distributions)**
>
> | Approach |  KL  | Fisher |
> |----------|------|--------|
> | BaM      | 0.07 |  0.55  |
> | ADVI     | 0.05 |  0.49  |
> | Flow     | 0.03 |  0.43  |
> | PoE      | 0.01 |  0.02  |
>
>
> **PosteriorDB: sesame**
>
> | Approach |  NLL  | Fisher |
> |----------|-------|--------|
> | BaM      | -5.52 |  27.56 |
> | ADVI     | -5.48 | 259.78 |
> | Flow     | -5.45 | 205.76 |
> | PoE      | -5.54 |   2.17 |
>
>
> **2. Computational cost and accuracy trade-off of the VI algorithm**
>
> > “Can the authors elaborate on the computational cost/complexity of computing the PoE within the BBVI framework?
> what is the tradeoff between compute time and posterior accuracy?
> can these metrics be reported for the evaluations in Section 4?”
>
> Having more experts will result in an improved posterior accuracy, as we show in the appendix in Figure D.1, but it will also increase the computational cost of solving for the weights. Once the experts are fixed, determining the PoE parameters $\alpha$ comes at a _relatively small cost_, since we need only solve a quadratic program to determine $\alpha$, which can be done in a matter of seconds, even in high dimensions.
>
> The timings of solving a QP with linear inequality constraints are below for different matrix dimensions $K$ (note that in this work, we only solve QPs up to size $K=100$):
> | $K$ | Mean time (s) |
> |---|-------------|
> | 10                      | 0.0007 |
> | 50                      | 0.0013 |
> | 100                    | 0.0043 |
> | 500                    | 0.0900 |
> | 1000                  | 0.3780 |
>
>
> The computational cost of the constrained least squares solve is $O(K^3)$; we note that this cost is not prohibitive for $K < 1000$, and for the problems in this paper, we did not require even this many randomly placed experts. For very large $K$, the cost will become prohibitive, and this provides motivation to work on better methods for expert selection in higher dimensions.
>
> In addition, the cost of computing scores is linear in the dimensionality of the latent variable $z$, and in many real-world problems, evaluating the unnormalizing target and its score is the dominating bottleneck.
>
> Finally, we are also happy to report additional metrics for the computational cost.
> We will revise this section of the paper to discuss the computational cost in more depth.
>
> **3. Expert placement: sensitivity, number of experts, and cost**
>
>
>  > “Even though it is listed as a limitation, more details on the initialization procedure would be appreciated:
> How sensitive are the presented results to a good initialization of the fixed collection of experts selected upfront?
> What is the number of experts used in the experiments, and how do results change with respect to varying it?
> What's the computational complexity of such initialization procedure?”
>
> **Sensitivity:** We place experts randomly on a hypercube around each mode, where the bounds are set according to the scale of the target. Thus, if the bounds of the hypercube are set incorrectly, then the tails of the target will not be modeled as accurately. We will add a discussion of this point to the paper.
>
> **Varying the number of experts:** In Appendix F.2, we describe the number of experts used (including the final number of “active” experts).
>
> **How to do results change with the number of experts:** We provide an empirical study in Appendix D.2. We show in Figure 2 the KL and Fisher divergences computed for K=10, 25, 50, 75, and 100. Indeed, the divergences decrease as we add more experts. In addition, we report the number of active experts in each case.
>
> **Computational cost:** With regard to the selection of experts, there is a _tradeoff_: if the experts are placed randomly, then we may require many, but the cost of each added expert is very small; on the other hand, with very carefully selected experts, we may only require a small number, but we will require a great deal more computation to place them effectively.
>
> We have explored the first regime in which the expert placement is extremely simple and cheap. The main bottleneck of the VI procedure is in the estimation of the weights, not in the expert procedure.
>
> We will revise the current manuscript to make this point about the cost of expert placement more explicit.

---

> ### Comment · Reviewer_TPhU · 2025-08-04
> **Thank you for the informative response!**
>
> I am happy to increase my score, as the provided rebuttal is informative and clarifies many of my raised concerns.
>
> I encourage the authors to make good use of the extra page to incorporate some of these insights, and to clearly point to the appendix for new or existing experiments that I had previously missed (sorry!)

---

> > ### Author Response · Authors · 2025-08-04
> >
> > Dear Reviewer,
> >
> > Thanks for the follow-up. We’re glad that the rebuttal cleared up many of your concerns. (I believe that authors can't see any edits to the reviews/scores yet, but we are happy to hear you're comfortable moving us to an "Accept".)
> >
> > We’ll use the extra page to report the new benchmarks and better highlight additional experiments and details in the appendix. Please see our response #2 to Reviewer xP2U on some planned changes for improving the discussion on the empirical results; we will also be sure to incorporate the other points (e.g., on computational cost of experts and estimation) we discussed above into the final revision.
> >
> > We appreciate the constructive discussion.
> >
> > Best,
> >
> > The authors

---

### Official Review · Reviewer_xP2U · 2025-07-03

**Clarity:** 3
**Significance:** 3
**Originality:** 4
**Rating:** 5
**Confidence:** 4

**Summary:**

This paper explores a new family of models for black box variational inference, where the models are a weighted product of experts, with each expert being a multivariate t-distribution. Noting a relationship to a Feynman identity of integrals from physics, the authors is how how this model can be written as a latent variable model, where the latent variable is Dirichlet distributed. This enable the authors to define a sampling strategy and a means to compute expectation values. From the standpoint of VI, the authors introduce a heuristic scheme for selecting experts, but also show how using a modifier Fisher divergence for the optimization of the weights of experts can lead to a tractable Quadratic program. A convergence theorem is also proven. On several synthetic experiments, the model is seen to work well.

**Questions:**

Could you move more information for both the heuristics for placing experts, and the details of the experiments into the main body? these two aspects seem under described in the text, and the paper would benefit from more details in the main body.

Do you have a possible path to move away from heuristics for selecting experts, and instead optimize their parameters?

Do you a have a notion of how big of a bias is introduced by the empirical estimate for the loss? Do you have any experiments that show if this significantly affects the optimization?

**Ethical Concerns:**

["NO or VERY MINOR ethics concerns only"]

**Limitations:**

yes

**Quality:**

3

**Strengths And Weaknesses:**

This is an interesting paper, introducing a new model and relying on mathematical tools from physics. The paper is clearly written, and includes description of the model, the sampling, the inference strategy, and the convergence.

One weakness is the method for selection of experts. This is essentially a heuristic method, that requires hill climbing and hand selecting means, inverse scale, and numbers of experts. Without a more automated and optimizable method for selecting experts and estimating their parameters, is is not clear how broadly useful this method can be, especially in anything but low dimensions of the data. Nonetheless, this work provides a good first step towards exploring this family of models.

The documentation of the experiments is also difficult to assess in the main text, although significantly more useful information is made available in the appendix.

---

> ### Author Rebuttal · Authors · 2025-07-31
>
> Thanks for your constructive feedback on the paper. We are happy to hear that you found the paper interesting, clearly written, and that you appreciated the overall approach. We respond to your comments and questions below.
>
> **1. Expert selection heuristics**
> > “One weakness is the method for selection of experts. This is essentially a heuristic method, that requires hill climbing and hand selecting means, inverse scale, and numbers of experts. Without a more automated and optimizable method for selecting experts and estimating their parameters, is is not clear how broadly useful this method can be, especially in anything but low dimensions of the data. Nonetheless, this work provides a good first step towards exploring this family of models.”
>
> With regard to the selection of experts, there is a _tradeoff_: if the experts are placed randomly, then we may require very many, but the cost of each added expert is very small; on the other hand, with very carefully selected experts, we may only require a small number, but we will require a great deal more computation to place them effectively.
>
> We have explored the first regime in which the expert placement is extremely simple and cheap, but as suggested by the reviewers, we are also looking at other places on this spectrum, where we consider more sophisticated and expensive ways to choose/adapt a smaller pool of experts (see #3 below for some strategies).
>
> **2. Requested additional details in main text**
>
> > “Could you move more information for both the heuristics for placing experts, and the details of the experiments into the main body? these two aspects seem under described in the text, and the paper would benefit from more details in the main body.”
>
> Yes, due to space, we moved many results and details to the experiments. With the extra page for the final revision, we will be able to better describe the expert placing and move more details of the experiments into the main body (e.g., details currently in Appendix F).
>
> In the main paper, we will revise the experiments section to provide a more complete overview of what to check out in the appendix, so the reader knows where to look, along with a _summary of the additional results in the appendix_, e.g.,:
> * empirical studies on expert placement and varying the number of experts,
> * understanding sensitivity to varying learning hyperparameters,
> * empirical verification of convergence theory, and
> * simulation studies on the normalizing constant estimation procedure.
>
> **3. Approaches beyond random expert placement**
>
> > “Do you have a possible path to move away from heuristics for selecting experts, and instead optimize their parameters?”
>
> Yes, there are many interesting directions that we are pursuing but did not make it into the paper for lack of space or being out of scope; here we chose to focus on other elements of the paper (e.g., studying the PoE family and theoretically characterizing convergence) and to use the simplest approach for expert selection. Below, we describe three approaches of increasing complexity.
>
> One idea is to use a score-based metric for finding where to place experts (in contrast to what we are currently doing, which involves placing a lot of experts randomly and pruning): that is, we check where the scores between the target and variational approximation are a poor fit, and place new experts to patch up the scores.
>
> Another approach is to select experts in a greedy “boosting” way (e.g., [1]) – that is, we iteratively repeat: 1) adding a new expert according to regions where there is large score mismatch, and then 2) running the described reweighting procedure to refit the weights of the experts.
>
> A final approach is that this family can be directly optimized using gradient descent. In addition, this family could be used in an amortized inference setting by letting the expert parameters and weights be a function of a shared set of parameters (over a data set). While we expect such approaches to utilize more compute, it could be interesting in settings where the user has a large computational budget for training, and would like posterior inference to be fast at test time.
>
> We are actively investigating all of these directions.
>
> [1] Miller et al. Variational boosting: iteratively refining posterior approximations. ICML, 2017.
>
> **4. Bias from empirical loss estimate**
>
> > “Do you a have a notion of how big of a bias is introduced by the empirical estimate for the loss? Do you have any experiments that show if this significantly affects the optimization?”
>
> The bias introduced due to sampling from $q(z| \alpha^{(t)})$ instead of $q(z| \alpha)$ does not prevent the iterates from converging to the optimal parameters. In particular, we prove in Theorem 3.1 that the iterates $\alpha^{(t)}$ obtained from solving Equation 27 converge to the optimal parameters (under no misspecification; otherwise, they converge to a neighborhood of the optimal parameters). We will add a sentence in this section clarifying this point.
>
> Finally, we also show experiments investigating the convergence under different values of $\eta_t,$ which controls the step size, and we find that the optimization is not very sensitive to this parameter.

---

> > ### Comment · Reviewer_xP2U · 2025-08-05
> >
> > Dear Authors,
> >
> > Thank you for your responses, and for agreeing to add the additional requested information. I continue to believe this is a interesting paper with solid technical developments.

---

### Official Review · Reviewer_qHA4 · 2025-07-04

**Clarity:** 3
**Significance:** 2
**Originality:** 4
**Rating:** 4
**Confidence:** 3

**Summary:**

This work introduces a new family of variational approximations where each member is a weighted product of experts and each expert is proportional to a multivariate t-distribution. Sampling from these distributions is non-trivial and this work presents a Feynman reparameterisation to tackle this issue. This reparameterisation reformulates these PoE distributions into latent variable models with auxiliary Dirichlet random variables. They then propose a Fisher Divergence based objective to fit this distribution to a given target distribution and prove that the corresponding updates converge to the optimal weights for the experts. They appraise this framework's performance by fitting synthetic datasets and datasets from posteriordb in Stan.

**Questions:**

For most of my primary concerns please refer to the weaknesses section above.

Two more questions I would like to ask -
1. What happens when you initialise the mixture components of a GMM at the right spots and only optimise the mixing proportion parameter? In terms of divergences, does the GMM not capture the target distribution accurately? How much better would a PoE be compared to this? When a GMM with a large number of components is initialised for the same target, does standard VI not infer a sparser GMM by driving down mixing proportion values for some of the components?
2.  From `Figure D.1` it seems that you have to always start with a very high number of experts, off which some are found to be redundant. Does this not lead to an increase in computational cost? Is there any way to be more efficient with this initialisation?

**Ethical Concerns:**

["NO or VERY MINOR ethics concerns only"]

**Final Justification:**

I've upgraded my rating from 3 to 4, please check our discussion below.

**Limitations:**

yes

**Quality:**

3

**Strengths And Weaknesses:**

**Strengths**
1. This work, to the best of my novel appears to be novel and original. The work seems inspired and geared towards solving present-day problems.
2. The theoretical foundations are solidly developed and clearly presented.

**Weaknesses**
1. Though this work motivates the development of this novel variational approximation from the perspective of having expressive and practical approximate posteriors, I do not see strong supporting evidence in the work to conclude either of these. Firstly, a limited evaluation has been conducted here and without more exhaustive testing on practical applications and uncertainty estimation benchmarks, it's hard to ascertain the quality of the posterior inferred.
2. The proposed approximation has very elegant foundations but, in the authors own words, optimising the parameters is `complicated` and without a strategy to place the experts at the right spots this approach cannot infer the right expert parameters. In this regard I would also like the authors to comment on the case where experts are placed at completely random locations. This seemingly non-negotiable requirement to correctly initialise the experts, shrinks the applicability of the proposed variational approximation.
3. According to `Result 2.1` text in `l 113- l 115 `, this PoE can be viewed as a continuous mixture of t-distributions. In this regard I would like the authors to comment on relevant approaches in Implicit variational inference [1][2] where continuous mixtures of Gaussians have been proposed as expressive variational approximations. How does the proposed approach compare to such formulations?


**Minor**
1. `l 25` >Each of such gradient evaluation..... . I think this line is a little vague. In my opinion, it really depends on the downstream task or purpose which determines whether a sample or a gradient is more useful. Kindly consider rephrasing this line or adding more information.
2.  It's hard to draw conclusions from the figures in the appendix, namely figure E.1 and F.2. In F.2 the legend is not readable.

[1] Unbiased Implicit Variational Inference, Titsias et. al. 2019
[2] Implicit Variational Inference for high dimensional posteriors, Uppal et. al. 2023

---

> ### Author Rebuttal · Authors · 2025-07-31
>
> Thank you for your time and your feedback on our paper. We are glad that you appreciated the “elegant foundations” and novelty of the Feynman representation of the product of experts, the clarity of the writing, and the theoretical grounding of this work, such as the auxiliary variable formulation and the analysis of the algorithm’s convergence.
>
> Below, we respond to your feedback and questions on the paper.
>
> **1. Expressiveness of the variational family**
>
> > “Though this work motivates the development of this novel variational approximation from the perspective of having expressive and practical approximate posteriors, I do not see strong supporting evidence in the work to conclude either of these.”
>
> We show that the family is expressive in several ways. First, the Feynman representation shows that the product of experts can be represented by a continuous location-scale mixture of t-distributions; indeed with just two experts, Figure 1 shows a variety of behaviors can be captured. Second, we show how the expressivity of the family impacts the convergence of the algorithm: in particular, the iterates converge to a neighborhood of the optimal weighting, where the radius of the neighborhood is determined by the amount of misspecification of the variational family (due to e.g., expert selection). Finally, we show how the PoE family can be used to model a variety of target distributions – we have added several new benchmarks in the rebuttal (see discussion below in #2).
>
> **2. Empirical results on posterior benchmarks**
>
> > “A limited evaluation has been conducted here and without more exhaustive testing on practical applications and uncertainty estimation benchmarks, it's hard to ascertain the quality of the posterior inferred.”
>
> Thanks for this feedback. We have now run the procedure on _several new target distributions_, including standard synthetic benchmarks and real-world data benchmarks from posteriordb. We believe that this addition adds to the diversity of targets studied in the empirical results. In particular, we have results covering the following targets:
>
> 1. Cross shape
> 2. Funnel
> 3. Skewed and heavy-tailed
> 4. Banana / Rosenbrock **(new)**
> 5. Multimodal, well-separated modes **(new)**
> 6. Multimodal, modes from mixing **(new)**
> 7. PosteriorDB - garch11: heavy-tailed + skewed
> 8. PosteriorDB - 8schools: funnel
> 9. PosteriorDB - gp-regr: light tailed and symmetric
> 10. PosteriorDB - sesame: **(new)**
>
> We believe this provides a range of target distributions that often arise in hierarchical modeling settings. The initial results for the new targets can be found in our response to Reviewer TPhU, and we plan to include additional posteriorDB models in the final version.
>
> **3. Expert placement at random locations**
>
> > “The proposed approximation has very elegant foundations but, in the authors own words, optimising the parameters is complicated and without a strategy to place the experts at the right spots this approach cannot infer the right expert parameters. In this regard I would also like the authors to comment on the case where experts are placed at completely random locations. This seemingly non-negotiable requirement to correctly initialise the experts, shrinks the applicability of the proposed variational approximation.”
>
> We understand that you are concerned about the need to place experts at the “right” locations. But it is not the case that there is “ground truth” expert placement and the model fails unless it recovers that ground truth.
>
> A concrete example is when the target is itself a product of t-experts – our expert placing procedure has no prior knowledge of the true locations, nor does it try to “infer” these locations. Instead, we place down a lot of experts and use the flexibility of reweighting to match the scores. We show that our procedure can still recover the target distribution without knowing or finding the true locations.
>
> **4. Relationship with implicit VI approaches**
>
> > “According to Result 2.1 text in l 113- l 115 , this PoE can be viewed as a continuous mixture of t-distributions. In this regard I would like the authors to comment on relevant approaches in Implicit variational inference [1][2] where continuous mixtures of Gaussians have been proposed as expressive variational approximations. How does the proposed approach compare to such formulations?”
>
> The implicit VI setting is an interesting one, and thanks for pointing us to these references. We will add a discussion of this setting and include additional relevant references to the updated paper.
>
> Via the Feynman representation and Result 2.1, the product of experts we consider can be viewed from the framework of implicit VI with a different kernel (t-distribution) and mixing measure (Dirichlet). In particular, the product of experts has a particular way of combining the auxiliary variables via Equations 15 and 9–10, whereas the typical implicit VI setting specifies a neural network with a mixing distribution over the parameters of that neural network.
>
> In this special case of continuously parameterized models, we are able to identify tools that lead to both an expressive family of densities but also a tractable normalizing constant via the Feyman representation. Our goal in this work has been to understand the potential and limitations of this special case.
>
> **5. Comparison with mixtures of Gaussians**
>
> > “What happens when you initialise the mixture components of a GMM at the right spots and only optimise the mixing proportion parameter? In terms of divergences, does the GMM not capture the target distribution accurately? How much better would a PoE be compared to this? When a GMM with a large number of components is initialised for the same target, does standard VI not infer a sparser GMM by driving down mixing proportion values for some of the components?”
>
> The weights in a mixture of experts play a different role than a product of experts. A useful way to distinguish between the two constructions is that mixture weights act like an “or,” whereas product weights act like an “and”. While the latter may seem restrictive, we also don’t require the experts themselves to be normalized, unlike in the mixtures case.
>
> Thus, we expect the two models to excel in different types of applications. To give an extreme and pedagogical example, if the target distribution is a student’s t distribution, we need an infinite mixture of Gaussians to model this.
>
> In addition, we note there are differences in _optimization_: products of experts are more amenable to score matching, as the objective is linear in the weights, leading to a constrained, nonnegative least squares (NNLS) problem. On the other hand, mixtures do not have efficient solutions for score-based divergences (or even the KL divergence), and are often optimized with gradient descent.
>
> These differences are important because the NNLS objective used in the PoE setting tends to learn _sparse_ solutions, so the parameters are driven down to 0. It’s unclear whether optimizing the GMM weights (e.g., using "standard VI" or ELBO optimization) would lead to a sparse solution.
>
> We will add a discussion to the revised paper highlighting the conceptual differences between these models.
>
> **5. Computational cost of using many experts**
>
> > “From Figure D.1 it seems that you have to always start with a very high number of experts, off which some are found to be redundant. Does this not lead to an increase in computational cost? Is there any way to be more efficient with this initialisation?”
>
> Using more experts increases the cost of determining the PoE parameter alpha, but this comes at a _relatively small cost_, since finding alpha boils down to solving a quadratic program, which can be done in a matter of seconds, even in high dimensions.
>
> We performed timings for solving a QP with linear inequality constraints for different matrix sizes $K$ (note that in this work, we only solve QPs up to size $K=100$):
>
> | K (matrix size) | Mean time (s) |
> |---|-------------|
> | 10                      | 0.0007 |
> | 50                      | 0.0013 |
> | 100                    | 0.0043 |
> | 500                    | 0.0900 |
> | 1000                  | 0.3780 |
>
> However, your question about other ways of initializing is an interesting one. We believe there are many ways this can be done, and for this paper we decided to focus on the simplest strategy. For future work, we have been actively investigating other strategies. See our response to Review xP2U for some concrete examples.
>
> **6. Additional clarity points**
>
> > “l 25 >Each of such gradient evaluation..... . I think this line is a little vague. In my opinion, it really depends on the downstream task or purpose which determines whether a sample or a gradient is more useful. Kindly consider rephrasing this line or adding more information.”
>
> Thanks for this feedback. Due to space limitations, we cut a contextual footnote related to this after the phrase “considerably more than what is provided”.
>
> The footnote said: “Here is a simple illustration of this point: suppose that $p$ is univariate Gaussian. Then one can determine the mean and variance of $p$ from the derivatives of $\log p$ at any two points in $\mathbb{R}^1$, whereas from $n$ samples (but no scores) one can only estimate these moments to $O(1/\sqrt{n})$ accuracy.”
>
> We are happy to add this back to the final revision of the paper.
>
> > “It's hard to draw conclusions from the figures in the appendix, namely figure E.1 and F.2. In F.2 the legend is not readable.”
>
> We will increase the sizes of the figures and the axis fonts.

---

> ### Comment · Reviewer_qHA4 · 2025-08-05
> **Reply to authors**
>
> I thank the authors for posting a detailed reply and individually addressing all my concerns. Please refer to the bullet points below for my response.
>
> - 1 and 2, thanks for adding more experiments. I am sure that these would go on to help readers who wish to develop approaches in this area or apply this one. Although with my comment about expressivity, I was also hinting at testing some higher-dimensional cases. Kindly help me with what is generally the dimensionality of the posterior you have tested your approach with.
>
> - 3, I am not fully convinced since this is not a solution rather a workaround that could in future have limitations. It's good to see that the cost of using a large number of experts is low. Thanks for adding that result.
>
> - 4, thanks, glad that this could improve your work. And thanks for explaining.
>
> - 5, there are two 5's above, so I am referring to the one about comparisons with GMMs.
>     >It’s unclear whether optimising the GMM weights (e.g., using "standard VI" or ELBO optimisation) would lead to a sparse solution.
>
>   I think you should investigate this, as it is quite relevant and the explanation is a bit vague. I am sure that there are sparse priors for GMMs that could enforce the same effect. I think this could improve your work.
>
> - 6 Thanks for accepting my suggestion.
>
> Overall, I thank the authors for their effort. I am not 100% convinced by the evaluation, but they have improved their manuscript, and I maintain that this work is original and interesting.

---

### Public Comment · ~Donlapark_Ponnoprat1 · 2026-03-06
**Normalizing constant**

Hi, and thanks for the paper!

I've noticed that the formula for the normalizing constant in Eq. (18) has $(1+\sigma^2(\omega))^{-\frac{\nu + D}{2}}$, while the one in the appendix Eq. (C.20) has $(1+\sigma^2(\omega))^{\frac{\nu}{2}}$ (which I think should be $(1+\sigma^2(\omega))^{-\frac{\nu}{2}}$). What is the correct formula?

---

### Note · Authors · 2025-08-11

We thank the AC and reviewers for their time and feedback, and recap our contributions and additions post-rebuttal.

We propose black-box VI with a _weighted product of t-experts_, which is a flexible family (e.g., skew, multimodality, & heavy tails). PoEs are rarely used in VI due to normalization and sampling challenges.

**Main contributions:**
* Using a Feynman identity, we recast the PoE as a continuous mixture of t’s **(Result 2.1)**; obtain a Dirichlet-expectation form of the normalizer **(Result 2.2)** and an auxiliary-variable sampler **(Result 2.3)**.
* An _algorithm_ to optimize this family: first we select a large set of experts and then reweight the experts by minimizing the Fisher divergence, which reduces to a _convex quadratic program_ that can be solved efficiently.
* **Thm 3.1** proves the algorithm converges exponentially fast to a neighborhood of the optimum under finite batches and variational family misspecification. The changing stochastic objective required specialized analysis (standard SGD analysis does not apply).
* **Empirical evaluation on synthetic & real targets:** Our approach performs better than Gaussian BBVI and competitive with or better than a flow.

Points addressed during rebuttal include:

**1. Expert selection & cost (Rev. qHA4, xP2U, UfQj, TPhU)**

We deliberately adopt a view reminiscent of a “basis set” approach, where you attempt to learn the coefficients of a basis expansion; here we begin with a large, expressive set of experts that is then reweighted.

Computational cost: selecting experts is cheap, and the cost is primarily in optimizing the weights, which comes at a relatively small cost via a quadratic program solve. In our response, we showed that the QP solver takes <1s for 1000 experts.

**2. Empirical evaluation (Rev. qHA4, TPhU)**

Our response included **new benchmarks**: more diverse targets, higher dimensions, and a total of 12 targets:
* Cross
* Funnel
* Skewed & heavy-tailed, with D=5, 20 **(new)**, 50 **(new)**
* Banana **(new)**
* Multimodal well-separated **(new)**
* Multimodal modes from mixing **(new)**
* PosteriorDB garch11
* PosteriorDB 8schools
* PosteriorDB gp-regr
* PosteriorDB sesame **(new)**

The revised paper, which will incorporate new experiments and expanded discussion, presents a novel PoE-based VI approach with theoretical guarantees and practical demonstrations, and it lays groundwork for future work with this PoE family, ranging from boosting VI to amortized inference.

---

### Decision · Program_Chairs · 2025-09-17

**Decision:**

Accept (spotlight)

**Comment:**

This paper introduces a neat trick: by leveraging the Feynman parameterization (that allows to re-write a product of denominators into an expectation w.r.t. a Dirichlet helper variable), the authors re-write a product of t-distributions (PoE) into an infinite Dirichlet-mixture over t-distributions. This trick is elegant and new, and shows promising results on a few inference tasks. The paper is very clearly written. All reviewers have voted for accepting the paper and the AC has also read large parts of the paper and fully agrees with the reviewers.

I recommend accepting the paper and assign it a spotlight (disclaimer to the authors: this is the AC's preliminary recommendation which might be changed by the SAC or the PC), as it introduces interesting, theoretical sound and strong method to treat PoEs.

Reasons why not give it a full oral (this was discussed among the reviewers) are that the trick is perhaps a bit niche (not clear whether it can lead to a broader theory) and the experimental evaluation which could have been a bit more ample.

Minor, Typo in the appendix: I think in C.10 it should be \Lambda instead of \Lambda^{-1}. The exposition in the appendix, leading to C.17, is also clearer than in the main paper. The authors should perhaps adopt it into the main paper in the revised version.